# MULTIPLE-PREDICTION-POWERED INFERENCE

**Charlie Cowen-Breen**[1][†]    **Alekh Agarwal**[2]    **Stephen Bates**[1]

**William W. Cohen**[3]    **Jacob Eisenstein**[3]    **Amir Globerson**[2]    **Adam Fisch**[3]

[1]Massachusetts Institute of Technology,  [2]Google Research,  [3]Google DeepMind
[†]Work done during internship at Google DeepMind.
ccbreen@mit.edu, fisch@google.com.

## ABSTRACT

Statistical estimation often involves tradeoffs between expensive, high-quality measurements and a variety of lower-quality proxies. We introduce Multiple-Prediction-Powered Inference (MultiPPI): a general framework for constructing statistically efficient estimates by optimally allocating resources across these diverse data sources. This work provides theoretical guarantees about the minimax optimality, finite-sample performance, and asymptotic normality of the MultiPPI estimator. Through experiments across three diverse large language model (LLM) evaluation scenarios, we show that MultiPPI consistently achieves lower estimation error than existing baselines. This advantage stems from its budget-adaptive allocation strategy, which strategically combines subsets of models by learning their complex cost and correlation structures.

## 1 INTRODUCTION

Efficiently estimating expectations of random variables under a fixed budget is a fundamental problem in many scientific settings. This paper addresses the general problem of optimally estimating such expectations when each random variable has an associated sampling cost, subject to a fixed total budget constraint. We are specifically motivated by the challenge presented by AI model evaluation, which is a critical, but often resource-intensive, step in model development and maintenance.

More concretely, in the AI model evaluation setting, a variable $X_1$ might represent a high-quality but expensive metric computed for every model response to an input query, such as a score from a human annotator or a powerful proprietary model used as an "autorater". The remaining variables, $X_2, \ldots, X_k$, might represent cheaper evaluation options (e.g., scores from smaller autoraters or rule-based systems), which can be viewed as covariates or proxies for the true score. Given the option to obtain samples of $X_1, \ldots, X_k$ (either jointly or independently), the primary objective is often to then estimate the mean of the high-quality score, $\mathbb{E}[X_1]$. In other cases, we may be interested in the mean difference between two scores, say, $\mathbb{E}[X_1 - X_2]$. The core difficulty in each case is in determining *which* of these variables to query, how *many times* to query them, and then finally how to *combine* them together to produce a statistically efficient, consistent estimate of the ground truth.

To formalize this, let $X := (X_1, \ldots, X_k)$ be a set of random variables with finite variance. We then consider the general problem of efficiently estimating any linear function of the mean of $X$ subject to a total observation budget $B$. That is, for some $a \in \mathbb{R}^k$, we want to estimate $\theta^* = a^\top \mathbb{E}[X]$ while spending no more than a total budget $B$ on collecting subsets of joint random variables $X_I = \{X_i\}_{i \in I}$ at cost $c_I$ for index subsets $I \subseteq \{1, \ldots, k\}$. More precisely, if $n_I$ is the number of times the subset $X_I$ is observed, we require that the $n_I$ satisfy a system of linear budget constraints of the form $\sum_I c_I n_I \leq B$, where the sum is over all such collected subsets $I$.

Estimating linear functions of $\mathbb{E}[X]$ allows for flexibility in how $\theta^*$ is defined. Given the AI evaluation setting above, for example, measuring $\mathbb{E}[X_1]$ corresponds to $a = (1, 0, \ldots, 0)$, while measuring $\mathbb{E}[X_1 - X_2]$ corresponds to $a = (1, -1, 0, \ldots, 0)$. The flexibility to observe subsets of $X$ also introduces a key trade-off that is unique with respect to previous related approaches to estimation. As we will show, observing variables jointly can be advantageous by reducing overall estimation variance.

This benefit, however, must be weighed against the data acquisition costs, $c_I$. We make no assumptions about the structure of these costs (e.g., they may be non-additive over the components in $I$). For instance, in our AI evaluation setting, obtaining predictions from multiple autoraters can often be parallelized, so the cost of multiple predictions (in latency) is not significantly more than that of the single slowest one. This is not always true; in medical diagnostics, for example, ordering many tests may become too taxing for a patient, and therefore undesireable or impossible to do jointly.

To solve this cost-optimal, multi-variate estimation problem, we introduce the Multiple-Prediction-Powered Inference (MultiPPI) estimator, which is a cost-aware generalization of the Efficient Prediction-Powered Inference (PPI++) estimator of Angelopoulos et al. (2023b), and extends it to **optimally leverage multiple types of predictions to power inference**. The MultiPPI estimator constructs a low-variance, consistent estimate of $\theta^\star$ by combining observations from judiciously chosen subsets of $X$. The core of our method is an optimization procedure that jointly determines the number of samples $n_I$ to draw from each subset $I$ and the corresponding linear weights $\lambda_I$ used to form the final estimate. We demonstrate that this allocation problem can be formulated as a second-order cone program (SOCP) for a single budget constraint, and a semidefinite program (SDP) for multiple budget constraints, and thus solved efficiently using standard techniques.

Theoretically, we show that the MultiPPI estimator is minimax optimal when the joint covariance matrix, $\Sigma = \text{Cov}(X)$, is known. For the typical case where it is unknown, however, we provide a framework for integrating an initial estimation phase where an approximation of the required covariance matrix, $\widehat{\Sigma}$, can be derived from either a small "burn-in" sample or a pre-existing labeled "transfer" dataset (a common scenario in applied settings)—and provide finite-sample bounds on the performance degradation that is incurred by substituting $\widehat{\Sigma}$ for $\Sigma$. Finally, we empirically demonstrate the effectiveness of this approach across three diverse LLM evaluation settings, including choosing between autoraters of different sizes, autoraters with different test-time reasoning configurations, and complex multi-autorater-debate scenarios. In all cases, our method achieves lower mean-squared error and tighter confidence intervals for a given annotation budget than existing baselines. We demonstrate that MultiPPI achieves this by automatically tailoring its strategy to the available budget $B$: that is, it learns to rely primarily on the cheaper autoraters when the budget is small, and naturally begins to incorporate more expensive, better autoraters as the budget increases. Taken together, our work provides a principled and computationally tractable framework for cost-effective, model-aided statistical inference, in settings with complex cost-versus-performance tradeoffs.

In summary, our main contributions are as follows:

- We introduce the **MultiPPI estimator** and frame the problem of finding the optimal subset sampling strategy and estimator weights as an efficient second-order cone program (SOCP).

- We prove that the MultiPPI estimator is **minimax optimal** when the covariance matrix $\Sigma$ of $X_1, \ldots, X_k$ is known, and provide finite-sample performance guarantees for the practical setting where the covariance matrix must first be estimated as a part of the overall inference problem.

- We demonstrate MultiPPI's applicability across multiple LLM evaluation settings, and show how it can effectively combine signals from different model sizes, reasoning configurations, and multi-agent debates to achieve **lower error and tighter confidence intervals** for a given budget.

## 2 RELATED WORK

Our work builds upon Prediction-Powered Inference (PPI; Angelopoulos et al., 2023a), a statistical framework for efficiently estimating population-level quantities by augmenting a small set of labeled data with predictions from a machine learning (ML) model. We specifically build on PPI++, the efficient extension of PPI introduced in Angelopoulos et al. (2023b), which also further improves variance by optimally reweighting these predictions. We describe PPI in greater depth in Section 3.

PPI is part of a broader class of statistical methods that leverage ML predictions for estimation. Its principles connect to classical control variates and difference estimators (Ripley, 1987; Särndal et al., 1992; Chaganty et al., 2018), which reduce variance by subtracting a correlated random variable with a known mean; the correlated variable in PPI is the ML prediction, whose mean can be (cheaply) estimated on unlabeled data. This approach also shares theoretical foundations with modern semiparametric inference, particularly methods from the causal inference literature like Augmented In-

verse Propensity Weighting (AIPW; Robins & Rotnitzky, 1995), Targeted Maximum Likelihood Estimation (TMLE; van der Laan & Rubin, 2006), and double machine learning (DML; Chernozhukov et al., 2018). Recently, PPI has been applied to Generative AI evaluation, where human annotations (or more generally, annotations from some trusted source) are combined with cheaper "autorater" outputs for efficient, unbiased estimates of model performance (Boyeau et al., 2024; Chatzi et al., 2024; Fisch et al., 2024; Angelopoulos et al., 2025; Saad-Falcon et al., 2024; Demirel et al., 2024).

Existing PPI frameworks, however, assume either a single predictor (Angelopoulos et al., 2023a;b) or a fixed set of predictors queried together (Miao et al., 2024). We address the common scenario where multiple predictors (e.g., autoraters) with different cost-performance profiles are available. This introduces a complex budget allocation problem: determining which predictors to query (individually, jointly, or in any joint subset), how often to query them, and how to combine the measurements they provide for a minimum-variance estimate under a fixed budget. Our work partially generalizes Angelopoulos et al. (2025), which optimizes a sampling policy for a single predictor. Unlike that work, however, which focuses on input-conditional policies and expected budget constraints, we find a fixed allocation policy that always satisfies a hard budget constraint for every run.

Our allocation problem is also related to budgeted regression with partially observed features (Cesa-Bianchi et al., 2011; Hazan & Koren, 2012) and active learning or testing (Settles, 2009; Kossen et al., 2021; Zhang & Chaudhuri, 2015). We emphasize, however, that our goal is estimation of a linear function of a population mean (i.e., $a^\top \mathbb{E}[X]$), and not regression (e.g., predicting $X_1$ from $X_2, \ldots, X_k$). While related, standard approaches to regression, including with partial observations, optimize for sample-wise predictive accuracy rather than for predictive accuracy of a population-level quantity. Our problem also connects to multi-armed bandit allocation for adaptive Monte Carlo estimation (Neufeld et al., 2014). A key difference is that these frameworks often use sequential, input-dependent policies to minimize regret, making it difficult to derive valid confidence intervals (CIs). Our framework, in contrast, computes a fixed allocation policy over predictive models (not individual inputs as in active learning or testing) and guarantees unbiased estimates with valid CIs. Even more broadly, our work shares similar high-level goals with transfer learning and domain adaptation (Pan & Yang, 2010; Ben-David et al., 2010, *inter alia*)—i.e., leveraging signals of varying quality and potential bias—though the statistical techniques are distinct.

## 3 PRELIMINARIES

In the following section, we introduce the general estimation problem of interest and summarize existing approaches. Suppose that we are interested in the mean of a random variable $X_1$, which is dependent upon another random variable $X_2$ (corresponding to estimating $a^\top \mathbb{E}[X]$ for $a = (1, 0)$ as described in §1). For example, in the AI model evaluation setting, $X_2$ may be an autorater's score for a model output to a user's query, and $X_1$ may be the ground truth quality of the response as measured by an expert human annotator. Suppose we have access to a small number ($n$) of i.i.d. samples that contain labels from both the target rater ($X_1$) and autorater ($X_2$), and a large number ($N$) of i.i.d. samples that contain only the autorater predictions ($\tilde{X}_2$). A naïve approach to estimating the mean is to simply take the sample average of $X_1$ and ignore $X_2$ entirely, which we denote by $\hat{\theta}_{\text{classic}} = \frac{1}{n} \sum_{j=1}^{n} X_1^{(j)}$. When the prediction $X_2$ is correlated with $X_1$ and easy to query, however, it is natural to consider the "prediction-powered" PPI estimator (Angelopoulos et al., 2023a;b):

$$\hat{\theta}_{\text{PPI}} = \frac{1}{n} \sum_{j=1}^{n} \left( X_1^{(j)} - X_2^{(j)} \right) + \frac{1}{N} \sum_{j=1}^{N} \tilde{X}_2^{(j)} \tag{1}$$

When we can afford to take $N$ to be very large, it is clear that the variance of $\hat{\theta}_{\text{PPI}}$ is much smaller than that of $\hat{\theta}_{\text{classic}}$ provided that our model predictions $X_2$ are close to $X_1$ in mean-squared error. When that fails, Angelopoulos et al. (2023b) propose adding a linear fit of the form:

$$\hat{\theta}_{\text{PPI++}} = \frac{1}{n} \sum_{j=1}^{n} \left( X_1^{(j)} - \lambda X_2^{(j)} \right) + \frac{1}{N} \sum_{j=1}^{N} \lambda \tilde{X}_2^{(j)}. \tag{2}$$

The parameter $\lambda$ may be chosen to minimize the variance of $\hat{\theta}_{\text{PPI++}}$ based on the observed labeled data. This strategy yields an estimator which asymptotically improves on $\hat{\theta}_{\text{classic}}$ and $\hat{\theta}_{\text{PPI}}$ in the

limit that $n \to \infty$ and $N \gg n$. Toward the setting where $n$ and $N$ may be comparable in size, if one is able to choose to or not to request a label $X_1^{(j)}$ for every observed unlabeled point $X_2^{(j)}$, a modification of $\hat{\theta}_{\text{PPI++}}$ allows one to do so in a cost-optimal way (Angelopoulos et al., 2025).

## 3.1 MULTIPLE PREDICTIVE MODELS

How should one adapt the preceding setting when one has access to many predictions, rather than just $X_2$? One option is to *stack* all predictions into a vector $X_{2:k} := (X_2, \dots, X_k)$ and choose $\lambda \in \mathbb{R}^{k-1}$ to be a vector in $\hat{\theta}_{\text{PPI++}}$; this is the estimator proposed by Miao et al. (2024), and can be written

$$\hat{\theta}_{\text{PPI++ vector}} = \frac{1}{n}\sum_{j=1}^{n}\left(X_1^{(j)} - \lambda^\top X_{2:k}\right) + \frac{1}{N}\sum_{j=1}^{N}\lambda^\top \tilde{X}_{2:k}^{(j)} \tag{3}$$

But this approach is suboptimal when (as is becoming standard) the best models may be available only for the highest prices: if any of $X_2, \dots, X_k$ is expensive to obtain, our ability to sample $X_{2:k}$ will be limited. This yields suboptimal results, as we show in §6. One may instead decide to perform PPI with just one model $X_i$, for whichever $i \neq 1$ has the best cost/accuracy tradeoff—but it is not clear *a priori* which one this is, or how much worse it may be compared to some combination of a cost-effective subset of $X$. Alternatively, perhaps it is possible for cheaper models be used to recursively estimate the means of more expensive models, thus creating a *PPI++ cascade*: for instance, if $k = 3$ and $(X_1, X_2, X_3)$ are in decreasing order of cost, we might consider

$$\hat{\theta}_{\text{PPI++ cascade}} = \frac{1}{n}\sum_{j=1}^{n}\left(X_1^{(j)} - \lambda X_2^{(j)}\right) + \frac{1}{N}\sum_{j=1}^{N}\left(\lambda \tilde{X}_2^{(j)} - \lambda' \tilde{X}_3^{(j)}\right) + \frac{1}{M}\sum_{j=1}^{M}\lambda' \tilde{\tilde{X}}_3^{(j)} \tag{4}$$

Each of these strategies can be realized as possible instances of the MultiPPI estimator we propose in the next section. Rather than coarsely limiting ourselves to sampling $X_{2:k} = (X_2, \dots, X_k)$ together, we allow the flexibility of sampling $X_I$ for generic index subsets $I \subseteq \{1, \dots, k\}$.

## 4 MULTIPLE-PREDICTION-POWERED INFERENCE (MULTIPPI)

As Section 3.1 highlights, it is not obvious how to best allocate a budget across a diverse suite of predictive models, where each model has its own cost and performance tradeoffs. We begin by defining the class of permissible estimators: We require that the number of times, $n_I$, that $X_I$ is sampled satisfies a linear budget constraint, specified by a set of non-negative costs $c_I \geq 0$ and total budget $B \geq 0$, for each index subset $I \subseteq \{1, \dots, k\}$.[1]

**Definition 1.** *An estimator $\hat{\theta}$ is **budget-satisfying** if it a measurable function of $n_I$ i.i.d. samples of $X_I$, for each $I \subseteq \{1, \dots, k\}$, such that $\sum_I n_I c_I \leq B$.*

To develop a principled search for the best budget-satisfying estimator, we begin by asking a simple question under idealized conditions:

**Question 1.** *If the covariance matrix, $\Sigma = \text{Cov}(X)$, is exactly known, what is the minimax optimal, budget-satisfying estimator of $\theta^* = a^\top \mu$ with respect to the mean-squared error, $\mathbb{E}[(\hat{\theta} - \theta^*)^2]$?*

The answer to Question 1 will provide us with a set of allocations $n_I$ and a corresponding budget-satisying estimator $\hat{\theta}_{\text{MultiPPI}}$ which we will evaluate on the $n_I$ samples of $X_I$, for each $I$. Once we have addressed this question, we address the case of unknown $\Sigma$ by describing strategies depending on the empirical covariance matrix $\hat{\Sigma}$, which may be estimated from data.

It turns out, perhaps surprisingly, that Question 1 reduces to the following tractable alternative:

**Question 2.** *If the covariance matrix, $\Sigma = \text{Cov}(X)$, is exactly known, what is the minimum variance, linear, unbiased budget-satisfying estimator of $\theta^* = a^\top \mu$?*

We demonstrate the equivalence of Question 1 and Question 2 in Theorem 2. For now, the "oracle" assumption on knowing the covariance matrix $\Sigma$ allows us to isolate the resource allocation problem

---

[1] In Section B, we extend the methodology to multiple budget constraints.

from the separate challenge of estimating how closely related $(X_1, \ldots, X_k)$ are to begin with, and to analyze what a good procedure for leveraging multiple predictive models under cost constraints should look like in theory. All proofs of our theoretical results are deferred to Appendix F.

## 4.1 MULTIPPI($\Sigma$): A MINIMAX OPTIMAL ALGORITHM

Recalling notation from Section 1, let $X \in \mathbb{R}^k$ denote a random vector of finite second moment with distribution $\mathbb{P}$. Let $\mathcal{I} \subseteq 2^{\{1,\ldots,k\}}$ denote a collection of index subsets which may be queried, and for any $I \in \mathcal{I}$, let $X_I = \{X_i\}_{i \in I}$ be the corresponding subset of $X$. Next, let $\underline{n} = \{n_I\}_{I \in \mathcal{I}}$, $n_I \in \mathbb{N}$ be an allocation of sample sizes, where $n_I$ i.i.d. samples are drawn for each subset $I$, and let $\underline{\lambda} = \{\lambda_I\}_{I \in \mathcal{I}}$, $\lambda_I \in \mathbb{R}^{|I|}$ define a corresponding set of weighting vectors for each subset $I$. Finally, let $\hat{\theta}(\underline{n}, \underline{\lambda})$ denote the weighted sum of sample means from each non-empty subset, i.e.,

$$\hat{\theta}(\underline{n}, \underline{\lambda}) = \sum_{I : n_I > 0} \frac{1}{n_I} \sum_{j=1}^{n_I} \lambda_I^\top X_I^{(j)}. \tag{5}$$

The MultiPPI estimator, $\hat{\theta}_{\text{MultiPPI}}$, is then defined as the optimal estimator in this class that minimizes the MSE subject to our unbiasedness (**U**) and budget (**B**) constraints:

$$\hat{\theta}_{\text{MultiPPI}} = \underset{\hat{\theta}(\underline{n}, \underline{\lambda})}{\arg\min} \; \mathbb{E}\left[ \left( \hat{\theta}(\underline{n}, \underline{\lambda}) - \theta^* \right)^2 \right] \quad \text{s.t.} \quad \textbf{U and B hold}, \tag{6}$$

where the constraints **U** and **B** are

$$\textbf{U} \iff \mathbb{E}[\hat{\theta}(\underline{n}, \underline{\lambda})] = \theta^* \text{ for all } \mathbb{P} \text{ of finite second moment} \quad \text{and} \quad \textbf{B} \iff \sum_I n_I c_I \leq B.$$

It can be shown that **U** reduces to a linear constraint on $\underline{\lambda}$, which makes our optimization convenient.

As previously discussed, the estimators of Equation (3) and Equation (4) can be viewed as special cases of this setup. For instance, Equation (3) corresponds to imposing the additional restriction that $\lambda_I = 0$ for all $I \in 2^{\{1,\ldots,k\}}$ except for $I = \{1, \ldots, k\}$ and $I = \{2, \ldots, k\}$; Equation (4) corresponds to the additional restriction that $\lambda_I = 0$ for all $I$ except for $\{1, 2\}$, $\{2, 3\}$ and $\{3\}$.

### 4.1.1 OPTIMIZATION

Solving Equation (6) is, in general, non-trivial. Since $\hat{\theta}(\underline{n}, \underline{\lambda})$ is linear in $X$, it can be shown that the optimal $(\underline{n}, \underline{\lambda})$ depend only on the covariance matrix $\Sigma$ of $X$, and so we will denote by $\hat{\theta}_{\text{MultiPPI}(\Sigma)}$ the solution to Equation (6) given any distribution such that $\Sigma = \text{Cov}(X)$. Then, it can be further shown (this follows from Theorem 2, presented next) that the MSE of $\hat{\theta}_{\text{MultiPPI}(\Sigma)}$ is

$$\mathcal{V}_B = \min_{\substack{\underline{n} \,:\, \textbf{B} \text{ holds} \\ \text{supp}(a) \subseteq \bigcup \{I : n_I > 0\}}} a^\top S(\underline{n}) a, \qquad S(\underline{n}) = \left( \sum_{I \in \mathcal{I}} n_I \Sigma_I^\dagger \right)^\dagger \tag{7}$$

where $\Sigma_I$ denotes the principle submatrix of $\Sigma$ on $I$, embedded back into $\mathbb{R}^{k \times k}$, and $\dagger$ denotes the Moore-Penrose pseudo-inverse.[2] The minimizing $\underline{n}$ of the above expression then also determines the optimal $\lambda_I$ to be the restriction of $n_I \Sigma_I^\dagger S(\underline{n}) a$ to the coordinates $I$. If the integrality constraints on $n_I$ are relaxed, we show in the appendix that this reduces to a second-order cone problem in the case of a single budget constraint, and a semi-definite program in the case of multiple budget constraints. This allows for Equation (7) to be solved efficiently using standard techniques (Section G).

### 4.1.2 MINIMAX OPTIMALITY

The minimal MSE $\mathcal{V}_B$ shown in Equation (7) has a more fundamental characterization. Here we show that it is in fact the minimax optimal MSE achievable by **any** budget-satisfying estimator, taken

---

[2]More formally, if $P_I \in \mathbb{R}^{k \times k}$ denotes the orthogonal projection onto $\text{span}(I) \subseteq \mathbb{R}^k$, we define $\Sigma_I = P_I \Sigma P_I^\top$, and so $\Sigma_I^\dagger := (P_I \Sigma P_I^\top)^\dagger$.

over the set of distributions $\mathcal{P}$ of covariance $\Sigma$. Consequently, the estimator defined by $\hat{\theta}_{\mathrm{MultiPPI}(\Sigma)}$ is minimax optimal over the set of distributions

$$\mathcal{P}_{\Sigma} = \{\text{distribution } P \text{ on } \mathbb{R}^k \colon \mathrm{Cov}(X) = \Sigma \text{ for } X \sim P\}.$$

Specifically, given costs $(c_I)_I$ and a budget $B$, let $\Theta_B$ denote the set of budget-satisfying estimators $\hat{\theta}$ per Theorem 1. We emphasize that we make **no** restriction on $\Theta_B$ to include only linear or unbiased estimators. Then the following result holds:

**Theorem 2** (Minimax optimality of MultiPPI for known $\Sigma$). *For all $\Sigma \succ 0$, we have*

$$\inf_{\hat{\theta} \in \Theta_B} \sup_{P \in \mathcal{P}_{\Sigma}} \mathbb{E}\left[(\hat{\theta} - \theta^*)^2\right] = \mathrm{Var}\left(\hat{\theta}_{\mathrm{MultiPPI}(\Sigma)}\right) = \mathcal{V}_B,$$

*where the variance is with respect to any distribution $P \in \mathcal{P}_{\Sigma}$.*

## 4.2 MultiPPI($\widehat{\Sigma}$): A practical algorithm

In practice, $\Sigma$ is rarely known and must be approximated by an estimated covariance matrix $\widehat{\Sigma}$. In general, there are many methods for constructing an estimate $\widehat{\Sigma}$ of $\Sigma$, and many of the theoretical properties of MultiPPI are agnostic to the particular choice made. The following theorem shows that, for any $\widehat{\Sigma}$ which converges in probability to $\Sigma$ as our budget tends to infinity, the MultiPPI estimator is asymptotically normal and achieves the optimal variance of Theorem 2. For this result, we need a technical condition which amounts to Equation (6) having a unique minimizer $\underline{n}$ as $B \to \infty$; we state it formally in Section F.2.

**Theorem 3.** *Suppose $X \in \mathbb{R}^k$ has finite second moment, and suppose that $\Sigma = \mathrm{Cov}(X)$ satisfies condition 14. Suppose that $\widehat{\Sigma} \overset{p}{\to} \Sigma$ in the operator norm as $B \to \infty$. Then for $\hat{\theta}_{\mathrm{MultiPPI}(\widehat{\Sigma})}$ arbitrarily dependent on any potential samples used to estimate $\widehat{\Sigma}$, we have*

$$\sqrt{B}\left(\hat{\theta}_{\mathrm{MultiPPI}(\widehat{\Sigma})} - \theta^*\right) \overset{d}{\to} \mathcal{N}(0, \mathcal{V}^*)$$

*as $B \to \infty$, where $\mathcal{V}^* = \lim_{B \to \infty} B\mathcal{V}_B$, and $\mathcal{V}_B$ is defined in Equation (7).*

It is important to note that the estimator $\hat{\theta}_{\mathrm{MultiPPI}(\widehat{\Sigma})}$ continues to enjoy unbiasedness, budget satisfaction, and asymptotic normality regardless of mis-specification in $\widehat{\Sigma}$.

A natural question concerns the level of suboptimality of $\hat{\theta}_{\mathrm{MultiPPI}(\widehat{\Sigma})}$ as a function of the degree of mis-specification of $\widehat{\Sigma}$ in finite samples. Below, we present a meta-result which serves to quantify the sensitivity of our procedure to errors in the specification of $\widehat{\Sigma}$.

**Theorem 4** (Stability of MultiPPI). *Let $P$ be a distribution of covariance $\Sigma$, and suppose that $\Sigma$ has minimum eigenvalue $\gamma_{\min}$. Let $\sigma^2_{\mathrm{classical}}$ denote the least MSE of any budget-satisfying sample mean of $\theta^*$. Let $\widehat{\Sigma}$ denote any non-random symmetric positive-definite matrix. Then we have*

$$\mathbb{E}\left[\left(\hat{\theta}_{\mathrm{MultiPPI}(\widehat{\Sigma})} - \theta^*\right)^2\right] \leq \mathcal{V}_B + \frac{4\sigma^2_{\mathrm{classical}}}{\gamma_{\min}}\|\widehat{\Sigma} - \Sigma\|_F.$$

*whenever $\|\widehat{\Sigma} - \Sigma\|_F \leq \gamma_{\min}/2$, where $\|\cdot\|_F$ denotes the Frobenius norm.*

In general, there are many methods for constructing an estimate $\widehat{\Sigma}$ of $\Sigma$, and Theorem 4 is agnostic to the particular choice made. In Section E.1, we show how to apply the meta-result above to derive a family of finite-sample bounds in a variety of distributional settings and for a variety of estimates $\widehat{\Sigma}$. In practice, we estimate $\Sigma$ from data, and find the Ledoit-Wolf estimator $\widehat{\Sigma}$ of the covariance matrix $\Sigma$ to perform best in our experiments. This is consistent with the fact that the Ledoit-Wolf estimate is designed to minimize $\mathbb{E}\|\widehat{\Sigma} - \Sigma\|_F$, and Theorem 4 shows that the error of $\hat{\theta}_{\mathrm{MultiPPI}(\widehat{\Sigma})}$ is controlled by $\|\widehat{\Sigma} - \Sigma\|_F$. In Theorem 9, we apply Theorem 4 to provide finite-sample performance guarantees on MultiPPI when the Ledoit-Wolf estimator is used to estimate covariance.

In our experiments, we evaluate $\hat{\theta}_{n, \lambda}$ on the same samples we used to estimate $\widehat{\Sigma}$. A similar approach was taken by Angelopoulos et al. (2023b) for PPI++ when tuning the value of $\lambda$, and we find that it is easy to implement and yields strong empirical results in practice.

Note that while the data reuse introduces bias in finite samples—due in part to the additional dependency of $\lambda_I$ on $X_I$ in Equation (5)—it preserves consistency and asymptotic normality in the limit as our budget $B$ and the number of (reused) burn-in samples tend to infinity. For an analysis of the bias introduced in finite samples, see Section E.5.

### 4.2.1 PROCEDURE

We now specify an easy-to-implement procedure that makes use of a burn-in of $N$ fully labeled samples to estimate $\widehat{\Sigma}$, and then also reuses the $N$ samples when estimating $\hat{\theta}_{\text{MultiPPI}(\widehat{\Sigma})}$. Specifically, we target the practical setting where we are given $N$ fully-labeled samples *a priori*, and have no ability to obtain more. This is typical of real-world settings in which we are given, or have already collected, a fixed dataset of "gold" labels that we are then trying to augment with PPI related techniques—and may be encapsulated by the budget constraint $n_{\{1,\dots,k\}} \leq N$. While we may not be able to obtain more fully-labeled samples, we may be afforded a separate computational budget for querying model predictions that then augment the $N$ fully-labeled samples; taken together, this setting is represented by a system of budget constraints.[3] In summary, we propose the following:

1. Estimate the covariance matrix $\widehat{\Sigma}$ on the $N$ fully-labeled samples, which we will reuse in Step 3.

2. Solve for the $n_I, \lambda_I$ which minimize Equation (6) given $\widehat{\Sigma}$. We refer to this as MultiAllocate($\widehat{\Sigma}$).

3. Sample the $n_I, \forall I \in \mathcal{I}$ additional data points accordingly, and return $\hat{\theta}_{\text{MultiPPI}(\widehat{\Sigma})}$, evaluated on both the $n_I$ additional data points for each $I \in \mathcal{I}$, and the $N$ (reused) initial samples.

## 5 EXPERIMENTAL SETUP

In each experiment, our goal is to estimate the mean $\theta^* = \mathbb{E}[X_1]$ of some random variable $X_1$ to be specified, which we will refer to as the *target*. This corresponds to the choice $a = (1, 0, \dots, 0)$ in our notation. We will also specify a *model family* $(X_2, \dots, X_k)$, together with a *cost structure* $(c_I)_{I \in \mathcal{I}}$. In each experiment, we are given some number of samples for which the entire vector $X = (X_1, \dots, X_k)$ is visible; we refer to such samples as *fully-labeled*. Given these samples, we perform the procedure outlined in Section 4.2.1: we estimate $\widehat{\Sigma}$ using these samples, sample from the auxiliary models $(X_2, \dots, X_k)$ according to the allocation specified by MultiAllocate($\widehat{\Sigma}$), and return $\hat{\theta}_{\text{MultiPPI}(\widehat{\Sigma})}$, evaluated on both the $N$ fully-labeled samples and the additional auxiliary data.

**Baselines:** In each experiment, we compare to several baselines. First, we compare to classical sampling. Second, we compare to PPI++ with each model included in the family (specified in Equation (2)), and to vector PPI++ with every model in the family (specified in Equation (3)).

**Experiment 1: Estimating Arena win-rates by autorater ensembles.** We focus on the Chatbot Arena dataset (Chiang et al., 2024), where of interest is the *win-rate* between a pair of models, which is the probability that a given user prefers the response of one model to that of the other. The randomness is taken over the prompt, the user, and the model responses. Here, we aim to estimate the win-rate between Claude-2.1 and GPT-4-1106-Preview; this is our *target*. Our *model family* consists of autoraters built on Gemini 2.5 Pro (without thinking) and Gemini 2.5 Flash. In our notation, we have $(X_1, X_2, X_3) = $ (human label, Gemini 2.5 Pro label, Gemini 2.5 Flash label). We draw model *costs* from the Gemini developer API pricing guide (Gemini API), see Section I. In this case, the cost of querying both models is simply the sum of the costs of querying each model independently.

**Experiment 2: Optimal test-time autorater scaling on ProcessBench.** In this experiment, we aim to estimate the fraction of correct solutions in the ProcessBench dataset (Zheng et al., 2024), given a small number of labeled examples. The task is simplified from its original form to a binary classification problem: determining whether a given math proof solution contains a process error, without identifying the specific step. We employ Gemini 2.5 Pro with a variable thinking budget as our autorater. Its accuracy correlates with the number of words expended in the thought, with performance gains saturating after approximately 500 words (see Figure 12 in the appendix). We create a family of four autoraters by checkpointing the model's thought process at 125, 250, 375,

---

[3]We explain how to solve the optimization problem posed by such systems in Appendix B.

and 500 words. A key aspect of this setup is the non-additive, cascading cost structure. Generating a response from a model with a larger thinking budget makes the outputs of all smaller-budget models available at a marginal cost. Consequently, the total cost for a subset of models S is modeled with two components: an input cost proportional to the sum of the word budgets in S, and an output cost proportional to the maximum word budget in S. Explicitly, for $S \subseteq \{125, 250, 375, 500\}$, we set

$$c_S = \texttt{output\_cost\_per\_word} \cdot \max S + \texttt{input\_cost\_per\_word} \cdot \sum S \quad (8)$$

For concision, in our results we refer to the model assessments after 125, 250, 375, and 500 words by "tiny," "small," "medium," and "large," respectively.

**Experiment 3: Hybrid factuality evaluation through multi-autorater debate.** Following Du et al. (2023), we evaluate the factual consistency of biographies for 524 computer scientists generated by Gemini 2.5 Pro. For each person $p \in \mathcal{P}$, we compare their Gemini-generated biography $b^p$ against a set of known grounding facts $\mathcal{F}^p = \{f_1^p, \ldots, f_{m_p}^p\}$ about the person. Our target metric is the proportion of *factually consistent pairs* $(b, f)$ within the total set $\mathcal{S} = \{(b^p, f^p) : p \in \mathcal{P}, f^p \in \mathcal{F}^p\}$. Concretely, we *target* the proportion $|\{(b, f) \in \mathcal{S} : (b, f) \text{ is factually consistent}\}| / |\mathcal{S}|$.

Ground-truth consistency of a pair $(b, f)$ is established by majority voting over five independent judgments from Gemini 2.5 Pro with thinking, a method validated by Du et al. (2023) to have over 95% agreement with human annotators. Our experiment, illustrated in Figure 13, assesses the performance of a more cost-effective model, Gemini 2.0 Flash Lite, as an autorater. To elicit better autoratings from queries to Gemini 2.0 Flash Lite, we bootstrap performance via multi-round debate. For a fixed number of agents $A \in \{1, 2, 3\}$, and a fixed number of maximum rounds $R \in \{1, 2\}$, we perform the following procedure: In each round, $A$ instances of Flash Lite are independently prompted to provide a reasoned judgment on the consistency of a pair $(b, f) \in \mathcal{S}$. A "pooler" instance of Flash Lite then consolidates these responses into a single *yes*, *no*, or *uncertain* output. A definitive *yes* or *no* concludes the process. If the pooler outputs *uncertain*, and the number of maximum rounds $R$ has not yet been reached, the $A$ agents review all prior responses and continue their debate in a new round. If the output remains *uncertain* after the final round, either *yes* or *no* is reported with equal probability—since the dataset is balanced, this outcome is fair insofar as it is as good as random guessing. We impose the maximum round restriction to encapsulate our budget constraint. For a given $(A, R)$, the cost is $A \cdot R$; for collections, the cost follows Equation (8).

## 6 EMPIRICAL RESULTS

We plot MultiPPI, and the baselines described in Section 5, for a budgets between 0 and $2k$ units of cost. We normalize model costs so that one unit of cost always represents exactly one query to our most expensive model. For each fixed budget and each method, we estimate the target, and construct asymptotic 95% confidence intervals $\mathcal{C}$ based on Theorem 3. We plot (i) coverage, $\mathbb{P}(\theta^* \in \mathcal{C})$; (ii) confidence interval width, $|\mathcal{C}|$; and (iii) mean-squared error $\mathbb{E}[(\hat{\theta} - \theta^*)^2]$. We report both the confidence interval width and the mean-squared error as a fraction of what classical sampling achieves (lower is better). In each case, the target is $\theta^* = \mathbb{E}[X_1]$, and $\mathbb{P}$ and $\mathbb{E}$ are computed with respect to the empirical distribution over the dataset observed (we perform $500k$ random trials with 250 given labels). Note that these 250 labeled points are evidently enough for all estimators considered to achieve good coverage (in Section D.2 we also include results with $1k$ labeled points). We implement the optimization scheme in `cvxpy`, and use CVXOPT as our choice of optimizer.

**Experiment 1: Chatbot Arena.** Results are shown in Figure 1 (top). Observe that different baselines dominate in different budget regimes. In the low-budget regime, scalar PPI++ with Gemini 2.5 Flash is the best baseline, while in the large-budget regime, vector PPI++ with both Gemini 2.5 Pro and Gemini 2.5 Flash is the best baseline. However, we see that MultiPPI improves on all baselines in all regimes. In the appendix, Figure 5 and Figure 2 plot the $\lambda_I$ and $n_I$ values learned by MultiPPI across budget regimes. Note that the learned values tend to the specifications for PPI++ with Gemini 2.5 Flash in the low-budget regime, and to the specifications for vector PPI++ in the large-budget regime, a finding that we rigorously prove happens in broader generality in Section E.2. Lastly, note that PPI++ with Gemini 2.5 Pro is suboptimal in all regimes. In other words, PPI++ with Gemini 2.5 Pro is not included in the Pareto frontier. This is because, for this task, its correlation with the label is no more than that of PPI++ with Gemini 2.5 Flash, yet it is strictly more expensive.

**Experiment 2: ProcessBench.** Results are shown in Figure 1 (middle). Again, we see that each baseline has a range of budgets for which it outperforms all other baselines. In particular, the cheaper models yield better performance when used in PPI++ in the smaller-budget regimes, while the more-expensive models yield better performance in the higher-budget regimes. In particular, vector PPI++, which uses all $k-1$ models, steadily improves as the budget increases, but only outperforms the other baselines at the highest budgets. This behavior is explained by Figure 12 in the appendix, which shows that predictive performance improves for larger thinking budgets. Thus the more expensive models yield higher correlation with the label and thus yield low-variance rectifiers; on the other hand, their high cost means that this decrease in rectifier variance is outweighed by our inability to draw an adequate number of samples from them in the low-budget regimes.

Of note is the fact that *the models which think for longer are not in general less biased*. This phenomenon is shown in Figure 15, which shows that thinking for longer is not enough to resolve the systematic bias present in the autorater. However, the figure also shows that simple debiasing schemes like PPI resolve this issue. Note that this trend is not reflected in the correlations between these models and the label, because correlation is invariant to addition of constants.

Finally, MultiPPI improves on all baselines methods in all regimes. Interestingly, Figure 3 and Figure 4 show that the parameters $\lambda_I$ and $n_I$ learned by MultiPPI transition from emulating PPI++ with the tiny model (which is the best baseline in the low-budget regime) to emulating a *cascaded version of PPI* (see Equation (4)), in which the medium model is used to debias the larger model.

**Experiment 3: Biography factuality evaluation.** Results are shown in Figure 1 (bottom). Once again, each baseline is dominant over the others in certain regimes; MultiPPI improves on all baselines in all regimes. Of note, however, is the fact that the coverage of all estimators considered, but MultiPPI and vector PPI++ in particular, degrades slightly in the large-budget regime (i.e., the 95% CI under-covers by $\approx 1\%$). We discuss this interesting phenomenon in Section E.5, and find that it does not occur when the number of labeled samples grows in constant proportion with the budget (see, for example, our additional results with $N = 1000$ fully-labeled samples in Section D.2).

In terms of the performance-vs-cost profile that MultiPPI leverages: Figure 13 shows that predictive performance increases, across many metrics, as the number of agents and number of rounds increases. Note, however, that a marginal increase in number of agents yields a greater increase in accuracy than a marginal increase in number of rounds (this is largely because the pooler is more likely to report "uncertain" after the end of the first round than after the end of the second; see Figure 14).

## 7    CONCLUSION

In this work, we introduce Multiple-Prediction-Powered Inference (MultiPPI), a framework for efficiently estimating expectations under budget constraints by optimally leveraging multiple information sources of varying costs. MultiPPI formulates the optimal allocation of queries across subsets of variables as a second-order cone program in the case of a single budget constraints, or a semi-definite program in the case of multiple—both can be efficiency solved using off-the-shelf tools. We provide theoretical guarantees, including minimax optimality when covariances are known, and demonstrate empirically across diverse LLM evaluation tasks that MultiPPI outperforms existing methods. By adaptively balancing cost and information, MultiPPI achieves lower error for a given budget, automatically shifting its strategy from cheaper proxies to more expensive, accurate predictors as the budget increases, thus offering a principled and practical solution for cost-effective inference.

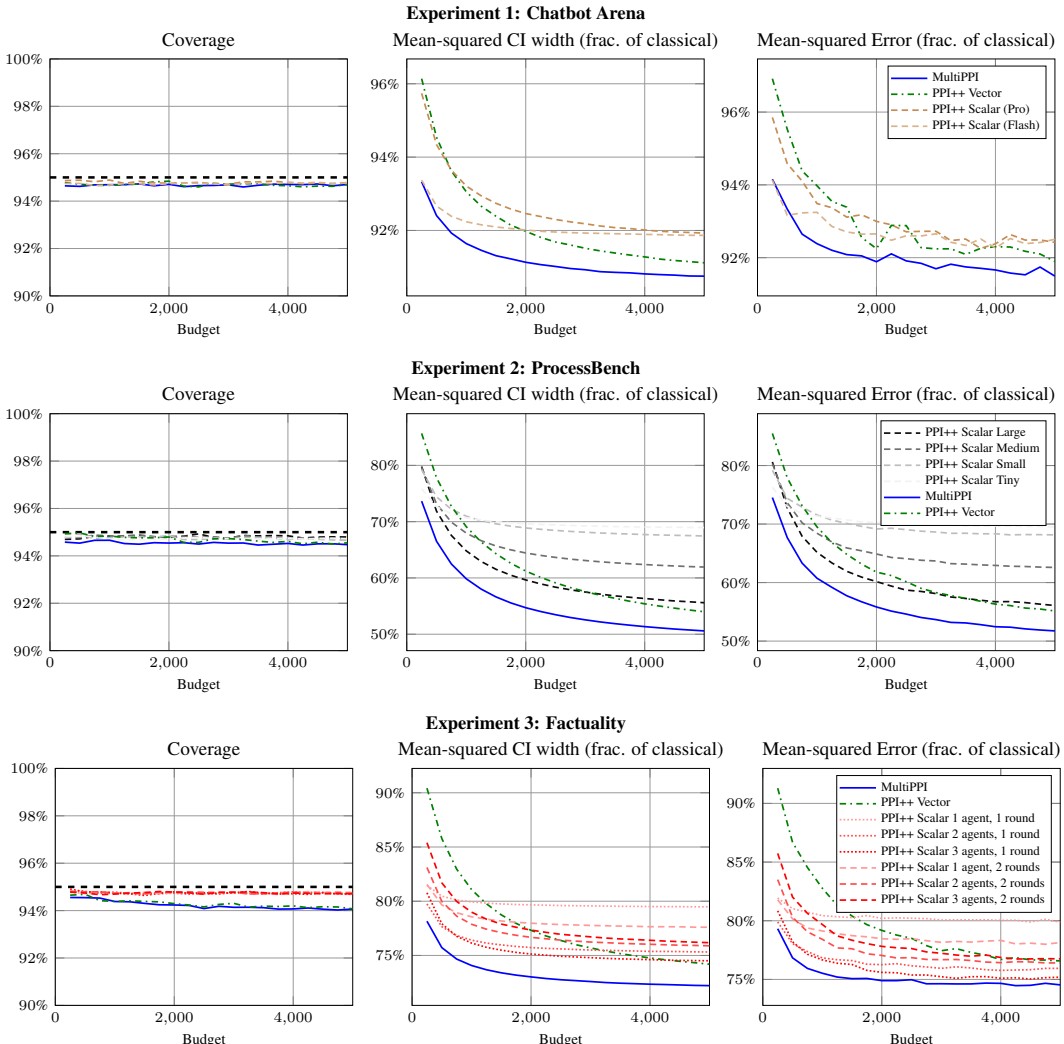

Figure 1: Results by budget for the experiments on Chatbot Arena (a), ProcessBench (b), and Factuality (c). For each estimator (all baselines and MultiPPI), the left column plots the empirical coverage of the 95% CI, the middle column plots the width of the 95% CI, and the right column plots the empirical mean-squared error of the point estimate. The fully-labeled sample size $N$ is 250.

## ACKNOWLEDGEMENTS

We are very grateful for insightful comments and suggestions by the anonymous reviewers. We are also grateful for helpful discussions with Jonathan Berant, Michael Beukman, Frederic Jørgensen, Petros Karypis, Chris Dyer, and Martin Wainwright.

## REPRODUCIBILITY STATEMENT

To ensure reproducibility, we provide a detailed specification of the algorithm in Section C. We also include implementation details in Section I, and address computational considerations in Section G. Finally, all experiments shown in §6 were averaged over 500k trials.

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

CONTENTS

## A  ETHICS STATEMENT

This paper describes fundamental research on techniques for constructing statistically efficient estimates of a target metric by optimally allocating resources across multiple types of proxy measurements. The primary intended use case which is analyzed in this work is the evaluation of generative AI systems, for which reliable evaluation is a core technical challenge. Efficient and precise estimates of model performance can help make AI systems easier to build, deploy, and monitor. We do not speculate about broader impacts that may follow from this technical contribution. Gemini was used for light copy-editing during the writing of this work.

## B  GENERALIZATION TO MULTIPLE BUDGET INEQUALITIES

We recall some notation. Fix a set $\mathcal{I}$ of index subsets $I \subseteq [k]$. For each $I \in \mathcal{I}$, let $c_I = (c_I^{(1)}, \ldots, c_I^{(m)}) \in \mathbb{R}_{\geq 0}^m$ denote the vector-valued cost of querying the collection of models indexed by $I$. Similarly, for each $I \in \mathcal{I}$, we let $n_I \geq 0$ be an integer denoting the number of times that the collection of models indexed by $I$ is queried. We let $\underline{n} = (n_I)_{I \in \mathcal{I}}$ refer to the associated allocation.

For a vector-valued budget $B \in \mathbb{R}_{\geq 0}^m$, we say that the allocation $\underline{n}$ satisfies the budget $B$, and write $\mathbf{B}(\underline{n}, B)$, if

$$\sum_{I \in \mathcal{I}} n_I c_I^{(1)} \leq B^{(1)}, \ldots, \sum_{I \in \mathcal{I}} n_I c_I^{(m)} \leq B^{(m)},$$

or more succinctly,

$$\sum_{I \in \mathcal{I}} n_I c_I \leq B.$$

Similarly, for each $I \in \mathcal{I}$, we let $\lambda_I \in \mathbb{R}^{|I|}$, and denote by $\underline{\lambda} = (\lambda_I)_{I \in \mathcal{I}}$ their collection. Let

$$\hat{\theta}_{\underline{n}, \underline{\lambda}} = \sum_{I \in \mathcal{I}: n_I > 0} \frac{1}{n_I} \sum_{j=1}^{n_I} \lambda_I^\top X_I^{(I,j)}$$

where $X^{(I,j)}$ denote independent copies of $X$ for every $I \in \mathcal{I}$ and $1 \leq j \leq n_I$. We say that the unbiased condition holds for $\underline{n}, \underline{\lambda}$, and write $\mathbf{U}$, if $\mathbb{E}\hat{\theta}_{\underline{n}, \underline{\lambda}} = a^\top \mathbb{E}X$ for every distribution of finite second-moment on $X$.

Note that the variance of $\hat{\theta}_{\underline{n}, \underline{\lambda}}$ depends only upon $\Sigma = \mathrm{Cov}(X)$. Thus we let

$$\hat{\theta}_{\mathrm{MultiPPI}(\Sigma)} := \hat{\theta}_{\underline{n}, \underline{\lambda}} \qquad \text{where } \underline{n}, \underline{\lambda} \text{ are chosen so that the resulting estimator has minimal variance under } \Sigma \text{ such that } \mathbf{B} \text{ and } \mathbf{U} \text{ hold.}$$

## C  DETAILED SPECIFICATION OF THE ALGORITHM

In this section, we outline the procedure used in all experiments in greater detail. First, we describe the algorithm for the case of a single budget inequality, for which a more-efficient procedure exists; second, we describe the general case, in which the procedure reduces to a semi-definite program (SDP). We first suppose that $\Sigma$ is known, and later explain the procedure in the case that it must be estimated from data.

### C.1  THE CASE OF A SINGLE BUDGET INEQUALITY, KNOWN $\Sigma$

We suppose that there is a random vector $X \in \mathbb{R}^k$ with known covariance $\Sigma$, and our goal is to estimate $\theta^* = a^\top \mathbb{E}X$ for some fixed $a \in \mathbb{R}^k$. There is some fixed collection $\mathcal{I}$ of index subsets $I \subset \{1, \ldots, k\}$ such that we may sample $X_I := (X_i)_{i \in I}$. We may sample $X_I$ a maximum of $n_I$ times, subject to the constraint that $\sum_{I \in \mathcal{I}} c_I n_I \leq B$ for some $c_I \geq 0$ and $B > 0$.

**Step 1:**  Solve the SOCP

$$\sup_{y \in \mathbb{R}^k} a^\top y \qquad \text{s.t.} \qquad \bigwedge_{I \in \mathcal{I}} \left\{ y_I^\top \Sigma_I y_I \leq c_I^{-1} \right\}$$

and obtain the solution $y_I^\star$ and the multipliers $\alpha_I^\star \geq 0$ for each $I \in \mathcal{I}$.

**Step 2:** Set

$$\lambda_I^\star = 2\alpha_I^\star \Sigma_I^{-1} y_I^\star$$

$$n_I^\star = \left\lfloor \left(\frac{B}{c_I}\right) \frac{\sqrt{c_I(\lambda_I^\star)^\top \Sigma_I \lambda_I^\star}}{\sum_{J \in \mathcal{I}} \sqrt{c_J(\lambda_I^\star)^\top \Sigma_J \lambda_J^\star}} \right\rfloor$$

for each $I \in \mathcal{I}$.

**Step 3:** For each $I \in \mathcal{I}$, independently sample $X_I$ $n_I^\star$ times, and compute the sample mean $\overline{\lambda_I^\star \cdot X_I}$. Return

$$\hat{\theta}_{\text{MultiPPI}(\Sigma)} = \sum_{I \in \mathcal{I}} \overline{\lambda_I^\star \cdot X_I}$$

with $(1 - \alpha)$-confidence intervals given by

$$\mathcal{C} = \hat{\theta}_{\text{MultiPPI}(\Sigma)} \pm z_{1-\alpha/2} \sqrt{\sum_{I \in \mathcal{I}} \frac{1}{n_I^\star} \widehat{\sigma^2}_{\lambda_I^\star \cdot X_I}}$$

where $\widehat{\sigma^2}_{\lambda_I^\star \cdot X_I}$ denotes the sample variance of $\lambda_I^\star \cdot X_I$, and $z_p$ denotes the $p^{\text{th}}$ quantile of the standard normal distribution.

## C.2 THE CASE OF MULTIPLE BUDGET INEQUALITIES, KNOWN $\Sigma$

We again suppose that there is a random vector $X \in \mathbb{R}^k$ with known covariance $\Sigma$, and our goal is to estimate $\theta^* = a^\top \mathbb{E} X$ for some fixed $a \in \mathbb{R}^k$. We may now sample $X_I$ a maximum of $n_I$ times, subject to the constraints that $\sum_{I \in \mathcal{I}} c_I^{(\ell)} n_I \leq B^{(\ell)}$ for some $c_I^{(\ell)} \geq 0$ and $B^{(\ell)} > 0$, with $1 \leq \ell \leq m$.

**Step 1:** Solve the SDP

$$\sup_{t \in \mathbb{R}} t \quad \text{s.t.} \quad \begin{pmatrix} \sum_{I \in \mathcal{I}} n_I P_I^\top \Sigma_I^{-1} P_I & a \\ a^\top & t \end{pmatrix} \succeq 0,$$

$$n_I \geq 0 \qquad\qquad \forall I \in \mathcal{I}$$

$$\sum_{I \in \mathcal{I}} c_I^{(\ell)} n_I \leq B^{(\ell)} \qquad\qquad \forall \ell \leq m$$

for real valued $n_I$, and obtain solutions $n_{I,\text{frac}}^\star$.

**Step 2:** Set

$$n_I^\star = \lfloor n_{I,\text{frac}}^\star \rfloor$$

$$\lambda_I^\star = n_I^\star \Sigma_I^{-1} P_I \left( \sum_{I \in \mathcal{I}} n_I^\star P_I^\top \Sigma_I^{-1} P_I \right)^\dagger a$$

for all $I \in \mathcal{I}$.

**Step 3:** As in the previous section, for each $I \in \mathcal{I}$, independently sample $X_I$ $n_I^\star$ times, and compute the sample mean $\overline{\lambda_I^\star \cdot X_I}$. Return

$$\hat{\theta}_{\text{MultiPPI}(\Sigma)} = \sum_{I \in \mathcal{I}} \overline{\lambda_I^\star \cdot X_I}$$

with $(1 - \alpha)$-confidence intervals given by

$$\mathcal{C} = \hat{\theta}_{\text{MultiPPI}(\Sigma)} \pm z_{1-\alpha/2} \sqrt{\sum_{I \in \mathcal{I}} \frac{1}{n_I^\star} \widehat{\sigma^2}_{\lambda_I^\star \cdot X_I}}$$

where $\widehat{\sigma^2}_{\lambda_I^\star \cdot X_I}$ denotes the sample variance of $\lambda_I^\star \cdot X_I$, and $z_p$ denotes the $p^{\text{th}}$ quantile of the standard normal distribution.

## C.3 THE CASE OF UNKNOWN $\Sigma$

In general, the approach is to construct an estimate $\widehat{\Sigma}$ of $\Sigma$ from data, and use this estimate for $\Sigma$ in the steps outlined above. In principle, it is possible to recycle the data used to construct $\widehat{\Sigma}$ in step 3 of the above procedures; this preserves asymptotic normality as a consequence of Theorem 3. Below, we detail one approach to doing this—the approach used in our experiments, and the approach outlined in Section 4.2.

Suppose that $a = (1, 0, \ldots, 0)$, and we have some hard limit $N$ on the number of samples available from $X_1$. This typically represents a "gold" label which is invaluable in some sense. We also suppose that these labeled samples are fully labeled—that is, that the entire vector $X = (X_1, \ldots, X_k)$ is visible in each case—or alternatively, that $N$ is small enough that they are relatively inexpensive to obtain model predictions for.

**Step 1:**   Construct the empirical covariance matrix $\widehat{\Sigma}$ from the $N$ fully-labeled samples.

**Step 2:**   Take $\mathcal{I}$ to be all subsets of models—that is, all subsets of $\{2, \ldots, k\}$—together with the set of all indices $\{1, \ldots, k\}$. Formally, $\mathcal{I} = \{\{1, \ldots, k\}\} \cup 2^{\{2, \ldots, k\}}$.

**Step 3:**   Run Section C.2 with any existing budget constraints, together with the constraint that $n_{\{1,\ldots,k\}} \leq N$, and obtain allocations $n_I^\star, \lambda_I^*$.

**Step 4:**   Sample accordingly, with the guarantee that the number of fully labeled samples $X_{\{1,\ldots,k\}}$ queried won't exceed the number available, $N$. These samples from step 1 may be reused for this.

**Step 5:**   Return the resulting estimator, as described in Section C.2.

# D ADDITIONAL EXPERIMENTS

## D.1 LEARNED ALLOCATIONS AND LINEAR PARAMETERS

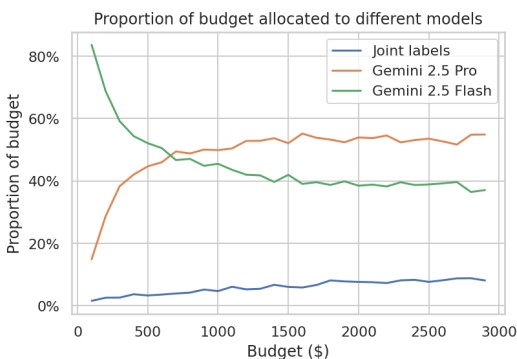

Figure 2: Proportion of budget allocated to different models in Experiment 1: ChatBot Arena. Gemini 2.5 Flash, the cheapest model, is most sampled in the low-budget regime, while the proportion of budget allocated to the joint (both models combined) increases monotonically with budget.

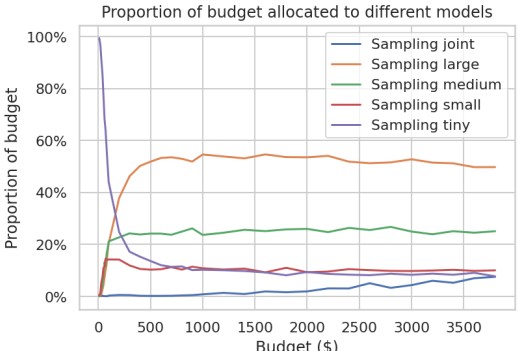

Figure 3: Proportion of budget allocated to different models in Experiment 2: ProcessBench. Tiny (125 word thinking budget) is most sampled in the low-budget regime, while the proportion of budget allocated to the joint (all models combined) increases monotonically with budget.

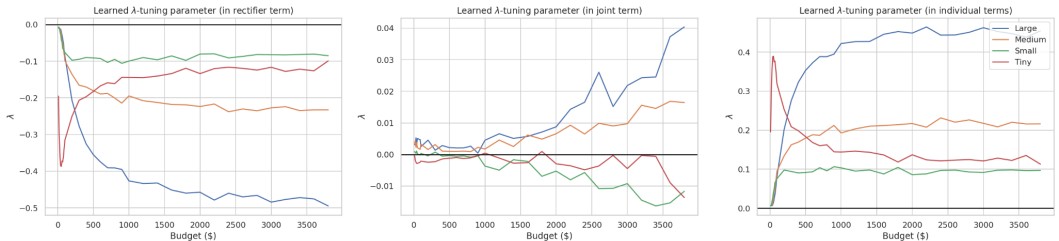

Figure 4: Linear parameters $\lambda_I$ learned across budget regimes in Experiment 2: ProcessBench. While only the tiny model (125 word thinking budget) has a nonzero linear parameter in the low-budget regime, a *cascading* behavior is learned in the large-budget regime: the cheaper models are prescribed the opposite sign from the more-expensive models in the joint term.

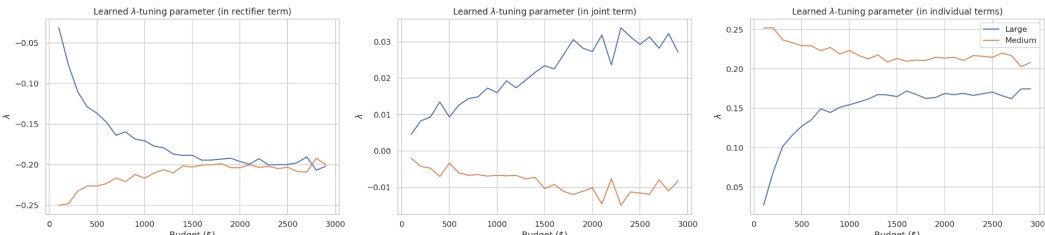

Figure 5: Linear parameters $\lambda_I$ learned across budget regimes in Experiment 1: ChatBot Arena. While only Gemini 2.5 Pro has a nonzero linear parameter in the low-budget regime, a *cascading* behavior is learned in the large-budget regime: the cheaper model (Gemini 2.5 Flash) is prescribed the opposite sign from the more-expensive model (Gemini 2.5 Pro) in the joint term.

### D.2 MULTIPPI BY VARYING NUMBER OF LABELED SAMPLES

In this section, we compare results of MultiPPI for a variable number of fully-labeled samples. MultiPPI continues to achieve smaller MSE than all baselines in all settings considered. In Figure 8, we plot the performance of MultiPPI with number of fully-labeled sample varying between $N = 10$ and $N = 200$. The extreme case $N = 10$ is shown in Figure 6, while $N = 1000$ is shown in Figure 7. Even for $N = 10$, we find that MultiPPI improves on all baselines in MSE. It is important to note that, for all methods, including the baselines, the coverage is significantly below 95% due to the small sample size. Nevertheless, even in this extreme setting, MultiPPI performs best in MSE.

### D.3 THE IMPACT OF SHRINKAGE COVARIANCE ESTIMATION

In this section, we discuss the impact of shrinkage covariance estimation on MultiPPI. We provide finite-sample bounds on the induced performance, and empirical results.

In Figure 9, we compare performance of MultiPPI with covariance estimation via (a) the empirical covariance matrix, and (b) the Ledoit-Wolf estimated covariance matrix.

For a finite-sample bound on our estimator with shrinkage estimation, see Theorem 9. For more general results on sensitivity to mis-specification, please refer to Theorem 4.

### D.4 SCALABILITY AND COMPUTATIONAL TRACTABILITY OF THE ESTIMATOR

SOCPs and SDPs are known to run in polynomial time in the number of contraints, which is, in our formulation, $|\mathcal{I}|$. In the preceding sections we have made the choice $\mathcal{I} = 2^{\{1,\ldots,k\}}$, but we show in this section that we may recover much of the same performance with a choice of $\mathcal{I} \subseteq 2^{\{1,\ldots,k\}}$ which grows only linearly in $k$. Specifically, we take $\mathcal{I} = \{\{1,\ldots,k\},\{2,\ldots,k\},\{2\},\ldots,\{k\}\}$, which corresponds to including terms for each model individually, as well as for their joint. We label the version of MultiPPI induced by this choice "MultiPPI (Restricted)." Figure 10 shows that the results of this method are very comparable to those of standard MultiPPI, in which we take $\mathcal{I}$ to be the collection of *all* subsets of $\{1,\ldots,k\}$. For more information on the computational tractability of the procedure, please see Section G. While our restricted formulation empirically recovers the performance of the full exponential search space by growing only linearly in $k$, a formal theoretical guarantee for this heuristic remains an open question. Proving that the restricted estimator incurs a small variance penalty in comparison to the full $O(2^k)$ formulation would validate its use for larger autorater ensembles.

### D.5 AUTORATER ACCURACY SCALING

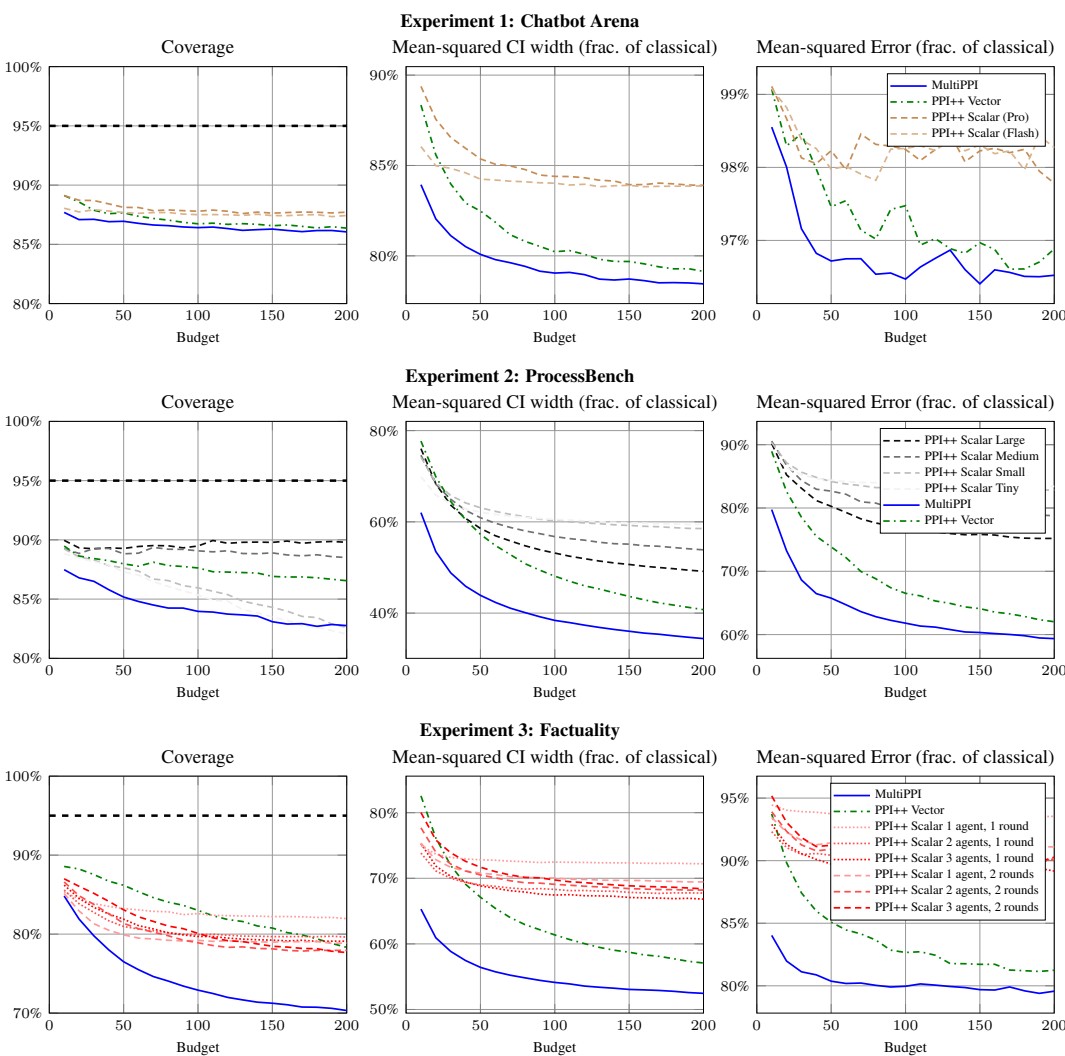

Figure 6: Results given only $N = 10$ labeled examples. Results are shown by budget for the experiments on Chatbot Arena (top), ProcessBench (middle), and Factuality (bottom). For each estimator (all baselines and MultiPPI), the left column plots the empirical coverage of the 95% CI, the middle column plots the width of the 95% CI, and the right column plots the empirical mean-squared error of the point estimate.

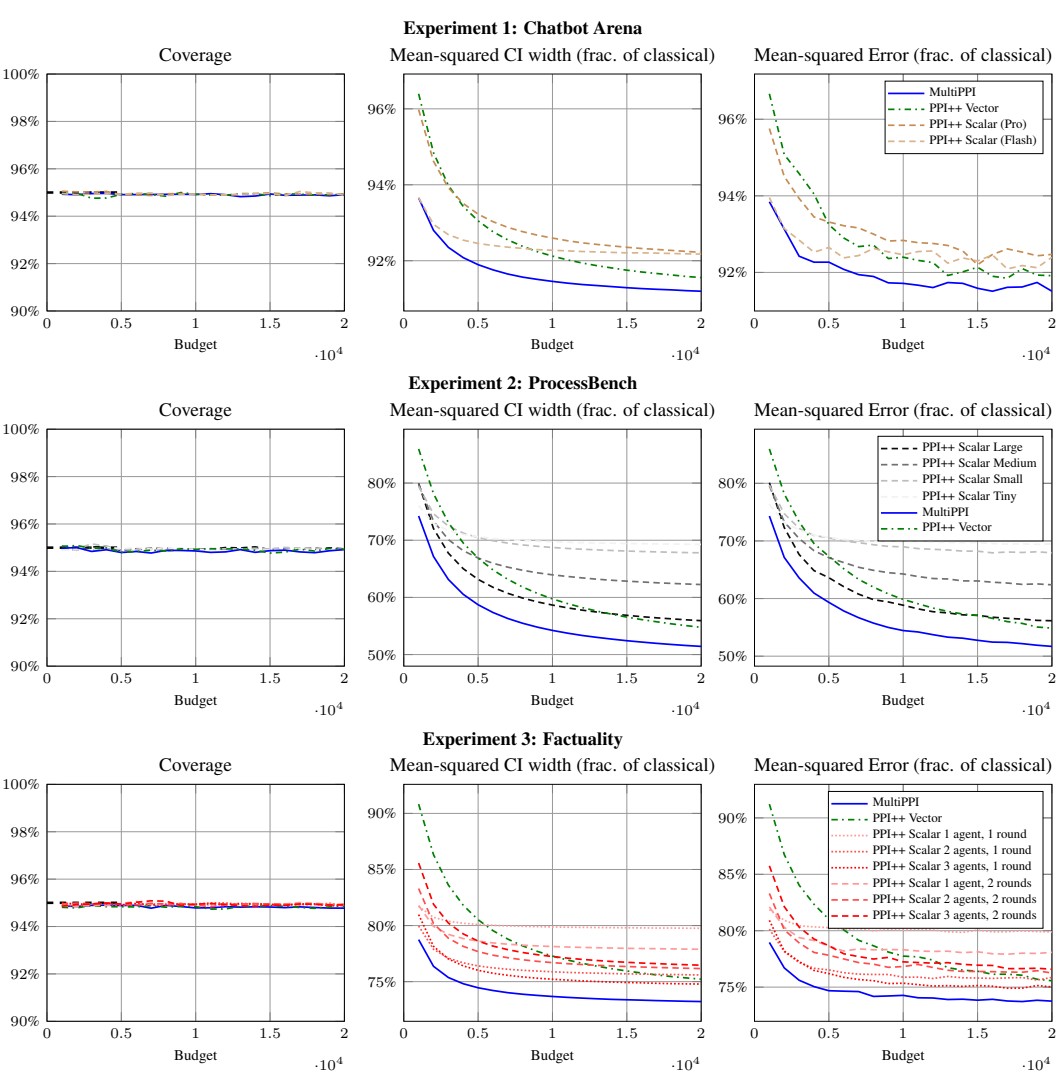

Figure 7: Results given $N = 1000$ labeled examples. Results are shown by budget for the experiments on Chatbot Arena (top), ProcessBench (middle), and Factuality (bottom). For each estimator (all baselines and MultiPPI), the left column plots the empirical coverage of the 95% CI, the middle column plots the width of the 95% CI, and the right column plots the empirical mean-squared error of the point estimate.

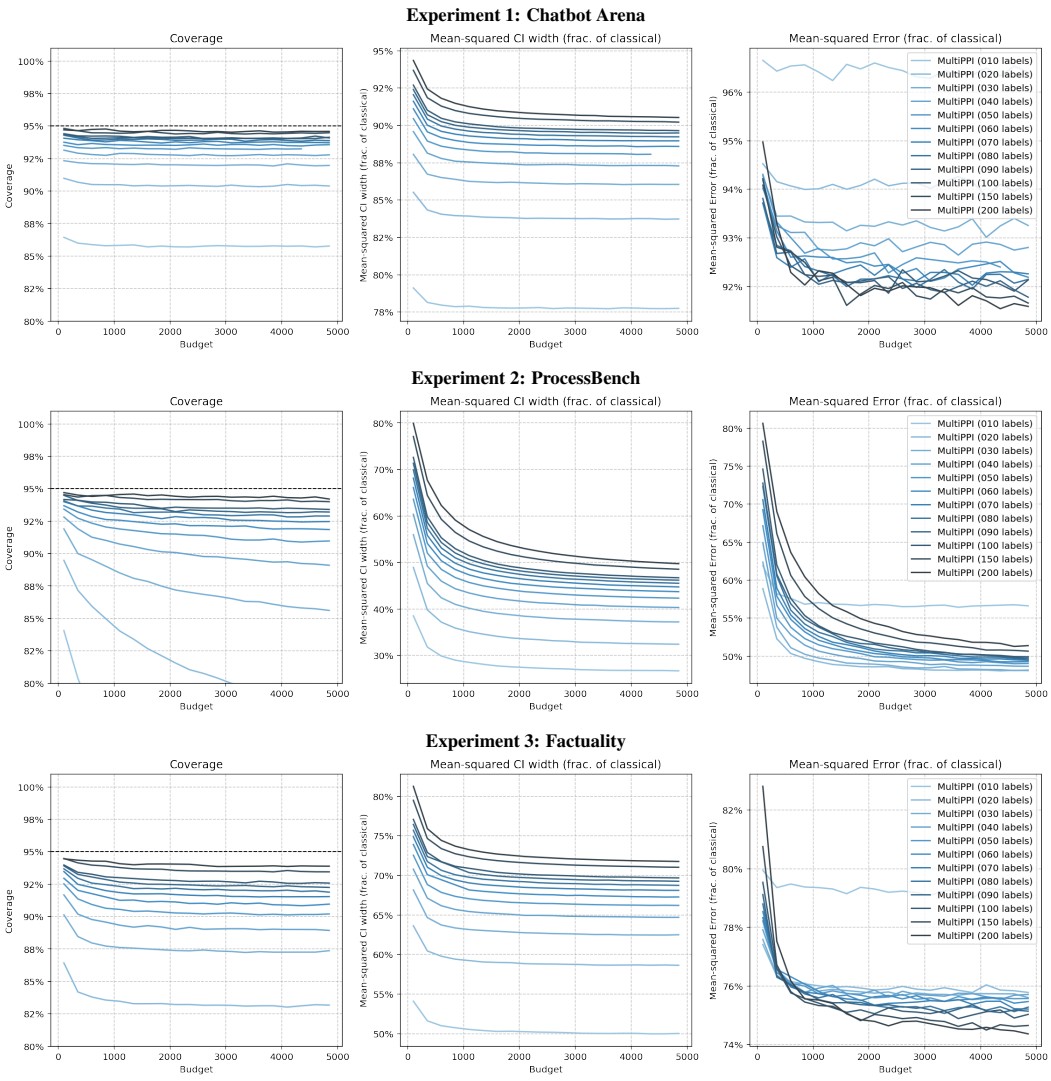

Figure 8: MultiPPI with varying number of fully labeled examples.

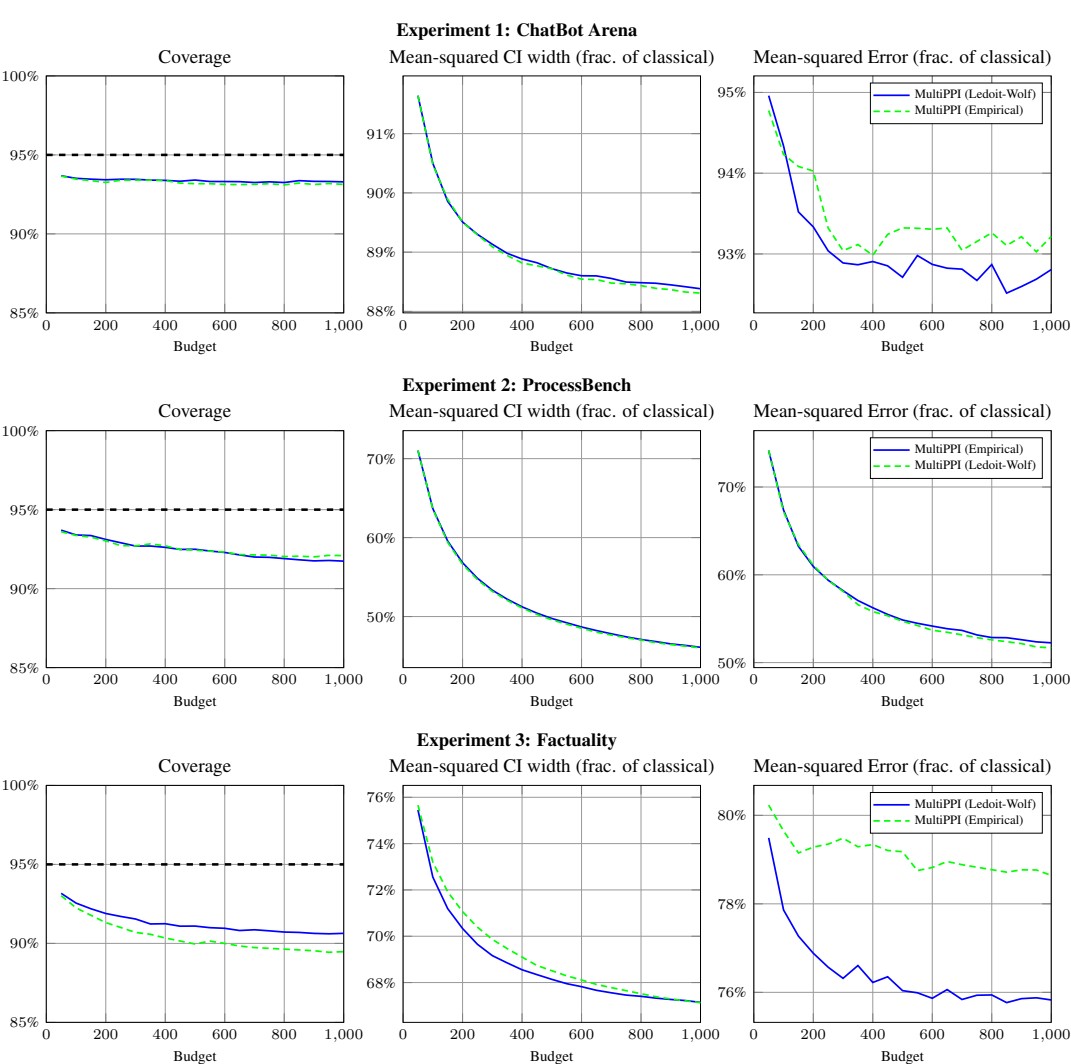

Figure 9: Comparison of results with different techniques for covariance estimation, for $N = 50$. We find that Ledoit-Wolf shrinkage covariance estimation yields best performance in all regimes.

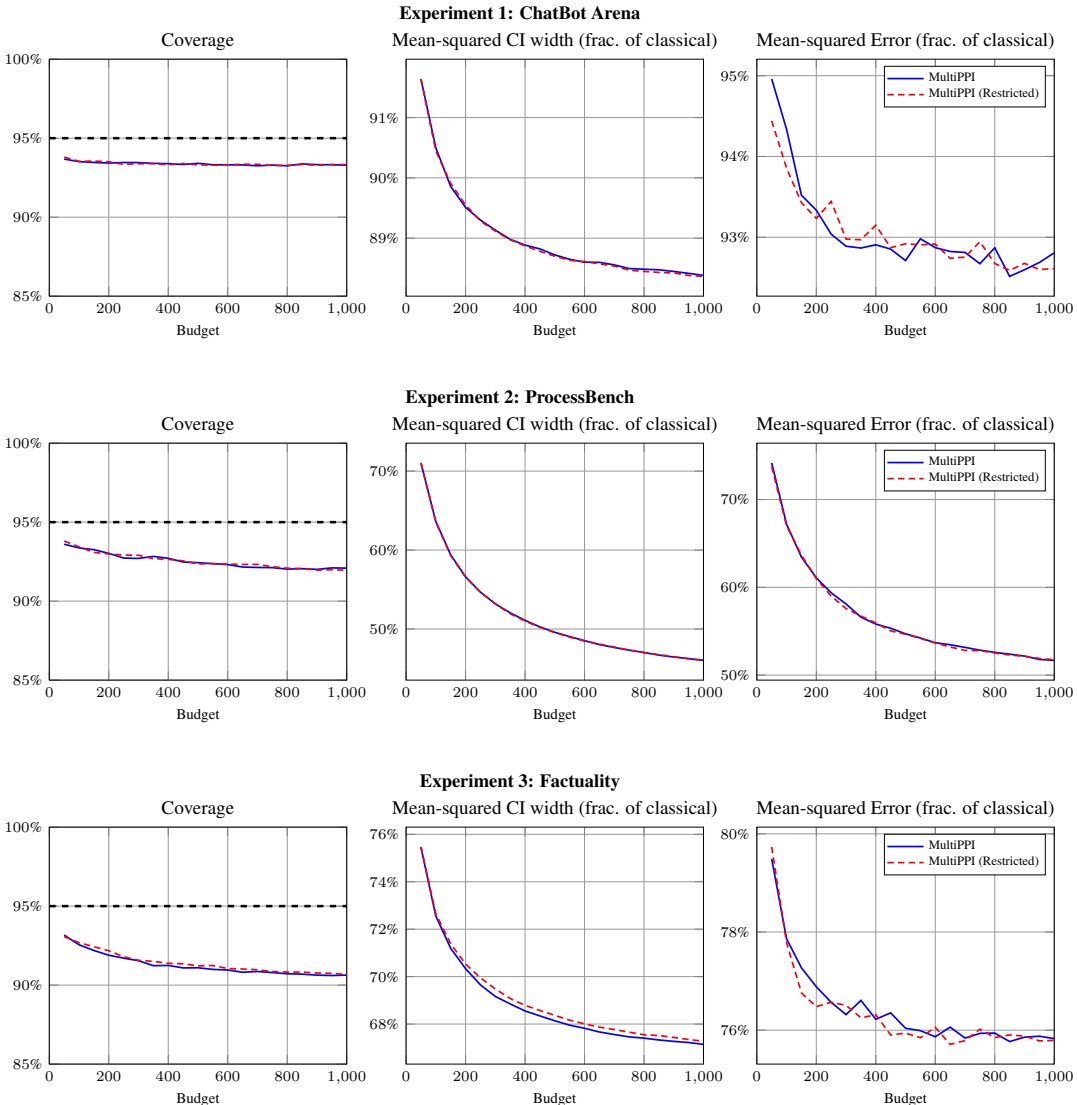

Figure 10: Comparison of MultiPPI for $\mathcal{I} = 2^{\{1,\dots,k\}}$ (default settings) with MultiPPI (Restricted), as defined in Section D.4.

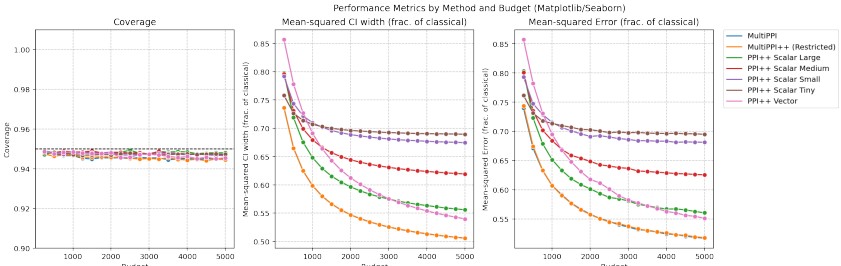

Figure 11: Performance at determination of process error vs. word budget. This is calculated via the procedure described in Appendix I. The majority of the improvement observed due to thinking occurs once 500 words of thought is reached, and plateaus around 1,000 words of thought.

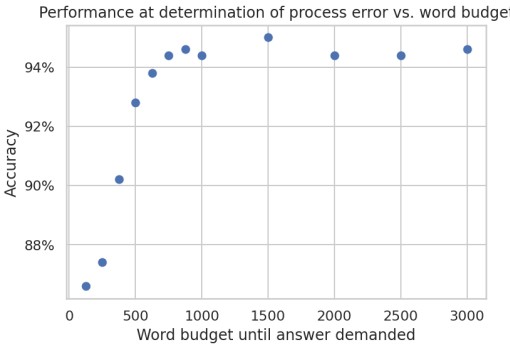

Figure 12: Performance at determination of process error vs. word budget. This is calculated via the procedure described in Appendix I. The majority of the improvement observed due to thinking occurs once 500 words of thought is reached, and plateaus around 1,000 words of thought.

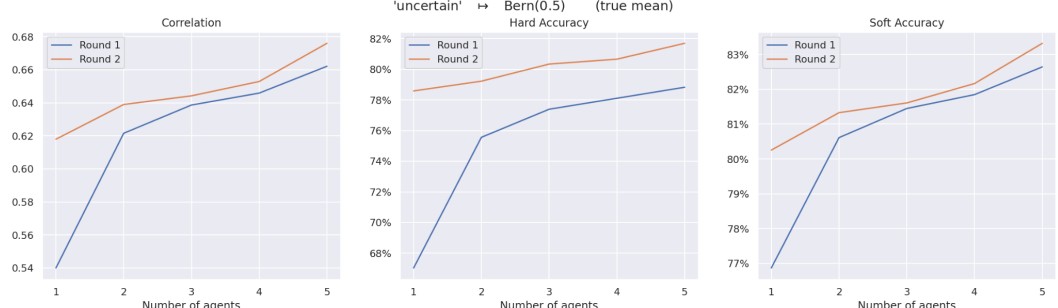

Figure 13: Performance at factuality evaluation with increasing number of agents and rounds of debate. Soft accuracy awards half a point to reporting an uncertain answer, while hard accuracy awards nothing.

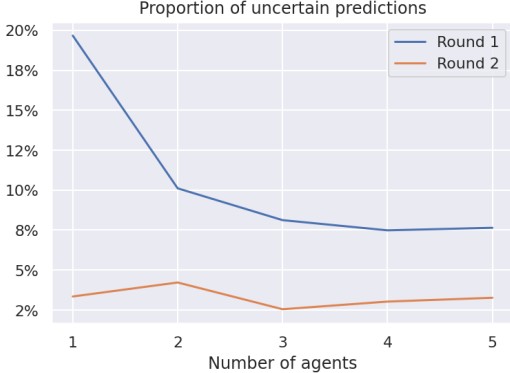

Figure 14: Proportion of uncertain predictions by number of agents and rounds of debate. An increased number of agents leads to fewer uncertain predictions, and almost all predictions are certain by the end of the second round of debate.

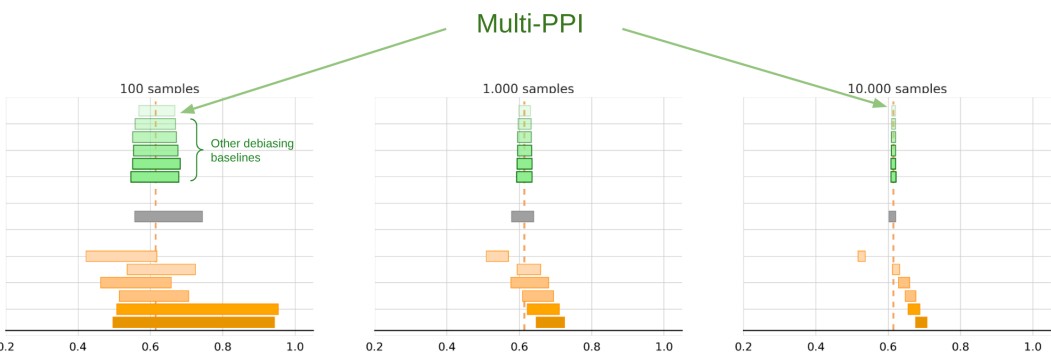

Figure 15: Different schemes for evaluation with autoraters on the ProcessBench dataset. Gray: classical sampling—no autoraters. Orange: pure autoraters, in decreasing order of thinking budget—note that the bias is increasingly pronounced with thinking budget. Green: various schemes for debiasing autoraters, including MultiPPI (top).

# E  ADDITIONAL THEORETICAL RESULTS

In this section, we provide additional theoretical guarantees on the MultiPPI estimator. First, we establish finite-sample bounds on the estimation risk, under bounded, sub-Gaussian, and autoregressive data distributions, and extend these results to the Ledoit-Wolf shrinkage estimator. Next, we characterize the behavior of the estimator at the boundaries of the budget spectrum. We prove that in the extreme low-budget limit, the allocation deterministically isolates the single model with the best correlation-to-cost ratio; conversely, in the large-budget regime, we show that the continuous relaxation of our discrete allocation problem is asymptotically optimal. Finally, we formally explain the empirically observed phenomenon of coverage decay, which occurs when the unlabeled budget grows while the labeled sample size remains fixed.

## E.1  FINITE-SAMPLE BOUNDS

We consider the setting of Appendix B, in which we may have several budget constraints. For the time being, we fix $a = (1, 0, \dots, 0)$ as in all experiments. Let $I^0 \in \mathcal{I}$ contain 1. A procedure which is similar to classical sampling is the following: Consider the choice $\underline{n}^0, \underline{\lambda}^0$ defined such that $n_I^0 = 0$ if $I \neq I^0$, and let $n_{I^0}^0$ be the maximal choice afforded by the budget (i.e. $n_{I^0}^0 = \max_{1 \leq \ell \leq m} \lfloor B^{(\ell)}/c_{I^0}^{(\ell)} \rfloor$). Then setting $\lambda_I^0 = 0$ if $I \neq I^0$, and $\lambda_{I^0}^0$ to be $a$ restricted to $I^0$, we recover the classical estimator

$$\frac{1}{n_{I^0}^0} \sum_{j=1}^{n_{I^0}^0} X_1^{(j)}$$

which has MSE $\sigma_1^2/n_{I^0}^0$, where $\sigma_1^2 = \Sigma_{11}$. We let $\sigma_{\text{classical}}^2 := \sigma_1^2/n_{I^0}^0$ denote this quantity.

We will compare $\hat{\theta}_{\text{MultiPPI}}$ to this in finite samples. Let $\widehat{\Sigma}_N$ denote the empirical covariance matrix constructed from $N$ i.i.d. samples from $P$, and let $\widehat{n}, \widehat{\lambda}$ denote the solution to MultiAllocate($\widehat{\Sigma}_N$), i.e. the minimizer of

$$\widehat{R}_N(\underline{n}, \underline{\lambda}) = \sum_{I \in \mathcal{I}: n_I > 0} \frac{1}{n_I} \lambda_I^\top \widehat{\Sigma}_N \lambda_I$$

such that **U** and **B** hold. On the other hand, let $\underline{n}^*, \underline{\lambda}^*$ denote the solution to MultiAllocate($\Sigma$), i.e. the minimizer of

$$R(\underline{n}, \underline{\lambda}) = \sum_{I \in \mathcal{I}: n_I > 0} \frac{1}{n_I} \lambda_I^\top \Sigma \lambda_I$$

such that **U** and **B** hold. In this section, we bound

$$R(\widehat{n}, \widehat{\lambda}) - R(\underline{n}^*, \underline{\lambda}^*).$$

**Theorem 5.** *Let $\gamma_{\min}$ denote the minimal eigenvalue of $\Sigma$, and $\delta = \|\Sigma - \widehat{\Sigma}_N\|_{op}$. Then for all $\delta \leq \gamma_{\min}/2$,*

$$R(\widehat{n}, \widehat{\lambda}) \leq R(\underline{n}^*, \underline{\lambda}^*) + 4\frac{\delta}{\gamma_{\min}} \cdot \sigma_{\text{classical}}^2$$

**Corollary 6.** *Suppose that $X_i \in [0, 1]$ almost surely. Then with high probability,*

$$R(\widehat{n}, \widehat{\lambda}) \leq R(\underline{n}^*, \underline{\lambda}^*) + c\left(\frac{\gamma_{\max}^{1/2}}{\gamma_{\min}}\sqrt{\frac{k \log k}{N}} + \frac{1}{\gamma_{\min}}\frac{k \log k}{N}\right)\sigma_{\text{classical}}^2$$

*for a universal constant c, and so*

$$\mathbb{E}R(\widehat{n}, \widehat{\lambda}) \leq R(\underline{n}^*, \underline{\lambda}^*) + c'\left(\frac{\gamma_{\max}^{1/2}}{\gamma_{\min}}\sqrt{\frac{k}{N}} + \frac{1}{\gamma_{\min}}\frac{k}{N}\right)\sigma_{\text{classical}}^2$$

*for another constant $c'$, where the expectation is taken over the $N$ labeled samples used to construct $\widehat{\Sigma}_N$.*

**Corollary 7.** *Suppose that $X$ is a subgaussian with variance proxy $K$. Then*

$$\mathbb{E}R(\widehat{n}, \widehat{\lambda}) \leq R(\underline{n}^*, \underline{\lambda}^*) + c'K^2 \left( \sqrt{\frac{k}{N}} + \frac{k}{N} \right) \sigma_{\text{classical}}^2$$

In the AR(1) model, and with bounded observations, choosing $N \gg k$ in the limit $k, N \to \infty$ is enough that $\mathbb{E}R(\widehat{n}, \widehat{\lambda}) \to R(\underline{n}^*, \underline{\lambda}^*)$. This follows as a special case of the following result.

**Corollary 8.** *Suppose, in addition to the conditions of Theorem 5, that $X_1, X_2, \ldots$ is a stochastic process such that $\mathrm{Var}\, X_t > c$ for all $t$, and $\mathrm{Corr}(X_t, X_s) \leq (1 - \rho)\rho^{|t-s|}$ for some $0 < c, \rho < 1$. Then we have*

$$\mathbb{E}R(\widehat{n}, \widehat{\lambda}) = R(\underline{n}, \underline{\lambda}) + o(1)$$

*whenever $k/N = o(1)$.*

Lastly, below we include a finite-sample bound for the performance of the estimator using Ledoit-Wolf shrinkage.

**Theorem 9** (Finite-sample bounds specialized to Ledoit-Wolf shrinkage)**.** *Let $\widehat{\Sigma}_N^{LW}$ denote the Ledoit-Wolf shrinkage estimator of $\Sigma$ based on $N$ samples. Let $\gamma_{\min}$ denote the minimum eigenvalue of $\Sigma$, and suppose that $X \in \mathbb{R}^k$ is sub-Gaussian with proxy $K$. Lastly, suppose that $\Sigma$ is not a multiple of the identity. Then for absolute constants $c_1, c_2$, we have*

$$\mathbb{E}\left[ \left( \hat{\theta}_{\mathrm{MultiPPI}(\widehat{\Sigma})} - \theta^* \right)^2 \right] \leq \mathcal{V}_B + \frac{4\sigma_{\text{classical}}^2}{\gamma_{\min}} \frac{1}{\sqrt{N}} \sqrt{c_1 K^4 \gamma_{\max}^2 k^2 + c_2 K^8 \gamma_{\max} k^3 / a^2}$$

*where $a^2 := \frac{1}{k} \left\| \Sigma - I \cdot \frac{\mathrm{tr}(\Sigma)}{k} \right\|_F^2$.*

For proofs, see Section F.4.

### E.2 BEHAVIOR OF THE ESTIMATOR IN THE LIMITING REGIMES

In this section, we explain a certain limiting behavior of the estimator in the regime of very low budget. Let $X = (X_1, \ldots, X_k)$ be a random vector of bounded second moment. We take $a = (1, 0, \ldots, 0)$, so that our target is $\mathbb{E}[X_1]$. We consider the setting (as is the case in all experiments) in which $\mathcal{I} = \{1, \ldots, k\} \cup \mathcal{I}_{\text{models}}$, where for each $I \in \mathcal{I}_{\text{models}}$ we have $1 \notin I$.

As in the experiments, we consider the budget model in which we have a fixed number of

For $I \in \mathcal{I}_{\text{models}}$, $\rho_I$ denote the multiple correlation coefficient of $X_I$ with $X_1$; that is, let $\rho_I = \mathrm{Cov}_I^\top \Sigma_I^{-1} \mathrm{Cov}_I$, where we define $\mathrm{Cov}_I := (\mathrm{Cov}(X_i, X_1))_{i \in I}$. The following result shows that, in the low-budget regime, MultiAllocate($\Sigma$) returns $n_I$ such that the only $I \in \mathcal{I}_{\text{models}}$ for which $n_I \neq 0$ is the one which minimizes the correlation/cost ratio $\rho_I / c_I$.

**Theorem 10.** *Fix $B > 0$ and consider the limit as $n_{[k]} \to \infty$. For each $I \in \mathcal{I}$, let $\alpha_I := \rho_I / c_I$. Suppose that $I^*$ uniquely minimizes $\alpha_I$ over $I \in \mathcal{I}_{\text{models}}$. Then the solution to MultiAllocate($\Sigma$) satisfies*

$$n_I \longrightarrow \frac{B}{c_I} \cdot \begin{cases} 1 & I = I^* \\ 0 & I \neq I^* \end{cases}$$

### E.3 ROUNDING IN THE LARGE BUDGET REGIME

In this section, we consider the suboptimality of the rounding scheme in the large budget regime. We consider the general setup in which we optimize

$$V_B(\underline{n}) = a^\top \left( \sum_I n_I P_I^\top \Sigma_I^{-1} P_I \right)^\dagger a \quad \text{s.t.} \quad n_I \geq 0, \sum_I c_I n_I \leq B, \mathrm{supp}(a) \subseteq \bigcup \{I : n_I > 0\}$$

We let $\underline{n}_{\text{frac}}^*$ denote the solution to this problem over all $\underline{n} \in \mathbb{R}_{\geq 0}^{|\mathcal{I}|}$, and $\underline{n}_{\text{int}}^*$ denote the solution over all $\underline{n} \in \mathbb{Z}_{\geq 0}^{|\mathcal{I}|}$. Let $\underline{n}_{\text{round}}$ denote the component-wise floor of $\underline{n}_{\text{frac}}^*$. Here we show that

$$\lim_{B \to \infty} \frac{V_B(\underline{n}_{\text{frac}})}{V_B(\underline{n}_{\text{int}}^*)} = 1$$

This follows from the fact that

$$V_B(\underline{n}^*_{\text{frac}}) \le V_B(\underline{n}^*_{\text{int}}) \le V_B(\underline{n}_{\text{round}})$$

and the limit $V_B(\underline{n}^*_{\text{frac}})/V_B(\underline{n}_{\text{round}}) \to 1$, to be proven next. Consider the difference vector $\underline{\delta} = \underline{n}^*_{\text{frac}} - \underline{n}_{\text{round}} \in [0,1]^{|\mathcal{I}|}$. Now observe that there is some $\underline{\nu}^* \in \mathbb{R}^{|\mathcal{I}|}_{\ge 0}$ such that

$$B V_B(\underline{n}^*_{\text{frac}}) = V_1(\underline{\nu}^*)$$

for all $B$, and equality holds if we take $\underline{n}^*_{\text{frac}} = B\underline{\nu}^*$. In particular, since we must have $\bigcup\{I : n^*_{\text{frac},I} > 0\} \supseteq \text{supp}(a)$, we may take the same to hold for $\underline{\nu}^*$. We therefore have

$$B V_B(\underline{n}_{\text{round}}) = B a^\top \left( B \sum_I \nu_I P_I^\top \Sigma_I^{-1} P_I + \sum_I \delta_I P_I^\top \Sigma_I^{-1} P_I \right)^\dagger a$$

$$= a^\top \left( \sum_I \nu_I P_I^\top \Sigma_I^{-1} P_I + \frac{1}{B} \sum_I \delta_I P_I^\top \Sigma_I^{-1} P_I \right)^\dagger a$$

Now since $\bigcup\{I : \nu_I^* > 0\} \supseteq \text{supp}(a)$, we may apply continuity of the inverse to conclude that

$$\lim_{B\to\infty} B V_B(\underline{n}_{\text{round}}) = a^\top \left( \sum_I \nu_I^* P_I^\top \Sigma_I^{-1} P_I \right)^\dagger a = V_1(\underline{\nu}^*)$$

and the limit is proven.

### E.4 THE CONTINUOUS PROBLEM AND ITS PROPERTIES

In this section, we study several properties of the the following continuous version of Equation (7), stated below:

$$\min_{\substack{\underline{\nu}\ge 0,\, \sum_I \nu_I c_I \le B \\ \text{supp}(a)\subseteq\{I:\nu_I>0\}}} a^\top S(\underline{\nu})a, \qquad S(\underline{\nu}) = \left( \sum_{I\in\mathcal{I}} n_I \Sigma_I^\dagger \right)^\dagger \tag{9}$$

where $\underline{\nu}$ may vary over $\mathbb{R}^{|\mathcal{I}|}$. Here, as is the case throughout, the inequalities $\underline{\nu} \ge 0$, $\sum_I \nu_I c_I \le B$ should be interpreted in the vector-valued sense (see Section B).

We are interested in the (set-valued) minimizers of Equation (9). More specifically, we are interested in the continuity of these minimizers in $\Sigma$, and how this relates to the discrete optimization problem of Equation (7). To address these topics, we introduce the notation, following Boyd & Vandenberghe (2004), that $\mathbf{S}^k_{++}$ denotes the set of symmetric positive definite $k \times k$ matrices. We are interested in the set-valued solution to the minimization problem

$$\underline{\nu}^*(A) = \operatorname*{argmin}_{\substack{\underline{\nu}:\underline{\nu}\ge 0 \\ \sum_I \nu_I c_I \le B_0 \\ \text{supp}(a)\subseteq\bigcup\{I:\nu_I>0\}}} a^\top \left( \sum_I \nu_I A_I^\dagger \right)^\dagger a \tag{10}$$

for $A \in \mathbf{S}^k_{++}$. More formally, recalling the set of feasible allocations $K = \{\underline{\nu} : \underline{\nu} \ge 0, \sum_I \nu_I c_I \le B\} \subseteq \mathbb{R}^{|\mathcal{I}|}$, we have

$$\begin{aligned} \underline{\nu}^* : \mathbf{S}^k_{++} &\rightrightarrows K \\ A &\mapsto \operatorname{argmin}_{\underline{\nu}\in K} F(\underline{\nu}, A) \end{aligned} \tag{11}$$

where

$$F(\underline{\nu}, A) = \begin{cases} a^\top \left( \sum_I \nu_I A_I^\dagger \right)^\dagger a & a \in \text{range}\left( \sum_I \nu_I A_I^\dagger \right) \\ \infty & \text{otherwise.} \end{cases} \tag{12}$$

We now connect the continuous problem to the discrete problem. Below we recall the discrete problem: define

$$\underline{n}_t^*(A) = \operatorname*{argmin}_{\substack{\underline{n}\in\mathbb{Z}^{|\mathcal{I}|}:\underline{n}\ge 0 \\ \sum_I n_I c_I \le tB_0 \\ \text{supp}(a)\subseteq\bigcup\{I:n_I>0\}}} a^\top \left( \sum_I n_I A_I^\dagger \right)^\dagger a \tag{13}$$

Then we have the following:

**Lemma 11.** *Suppose that $\underline{\nu}^*(A^*) = \{\underline{\nu}^0\}$ is a singleton and that $A^{(N)} \to A^*$. Then for every $\epsilon > 0$, there is some $N_1$ so that*

$$\left\| \frac{\underline{n}}{t} - \underline{\nu}^0 \right\| < \epsilon$$

*whenever $\underline{n} \in \underline{n}_t^*(A^{(N)})$ and $N, t \geq N_1$.*

For the proofs, see Section F.

### E.5 DECAY OF COVERAGE IN THE LARGE BUDGET REGIME

In this section, we discuss the phenomenon of decaying coverage as $B \to \infty$. Note that this is not unique to MultiPPI: it can be seen occuring to all baselines we compare to, and is especially pronounced for PPI++ vector. After discussing the phenomenon, we describe one way to avoid it.

Since, to the best of our knowledge, this phenomenon has not been observed in other works concerning PPI++, we focus our discussion on the PPI++ estimator and explain why it happens in that setting. Recall from Equation (2) the PPI++ estimator

$$\hat{\theta}_{\text{PPI++}} = \frac{1}{n} \sum_{i=1}^n \left( Y_i - \widehat{\lambda} X_i \right) + \frac{1}{N} \sum_{j=1}^N \widehat{\lambda} \tilde{X}_j$$

where $\{(X_i, Y_i)\}_{i \leq n}$ are i.i.d. according to some joint distribution $\mathbb{P}$, and $\{\tilde{X}_j\}_{j \leq N}$ are i.i.d. $\mathbb{P}_X$.

Angelopoulos et al. (2023b) (as well as many works before, in the context of control variates) propose a choice of $\widehat{\lambda}$ which depends on $\{(X_i, Y_i)\}_{i \leq n}$; namely, they let

$$\widehat{\lambda} = \frac{N}{n+N} \frac{\widehat{\text{Cov}}(X_{1:n}, Y_{1:n})}{\widehat{\text{Var}}(X_{1:n})}$$

where $\widehat{\text{Cov}}(X_{1:n}, Y_{1:n})$ and $\widehat{\text{Var}}(X_{1:n})$ are the relevant empirical covariance and variance computed from $\{(X_i, Y_i)\}_{i \leq n}$. This choice introduces bias in finite samples, and MultiPPI exhibits a similar behavior, as discussed in Section 4. In the limit theorems provided in this work, c.f. Theorem 3, and in Angelopoulos et al. (2023b), it is assumed that the number of labeled samples (here, denoted $n$) tends to infinity. But this is not the situation presented in our experimental results.

Here we consider the bias of $\hat{\theta}_{\text{PPI++}}$ for fixed $n$ as $N \to \infty$. This bias is exactly

$$\text{bias}(\hat{\theta}_{\text{PPI++}}) := \left| \mathbb{E}[\hat{\theta}_{\text{PPI++}}] - \mathbb{E}[Y] \right| = \left| \mathbb{E}[\widehat{\lambda}(X_1 - \tilde{X}_1)] \right| = \frac{N}{n+N} \left| \text{Cov}\left( X_1, \frac{\widehat{\text{Cov}}(X_{1:n}, Y_{1:n})}{\widehat{\text{Var}}(X_{1:n})} \right) \right|$$

by independence of $\widehat{\lambda}$ and $\tilde{X}_1$. Now for fixed $n$, and $N \to \infty$, the right-hand side converges upward precisely to the covariance of $X_1$ with the sample regression slope of $Y$ onto $X$, which is not in general zero. Therefore, the bias will increase but stay bounded as $N \to \infty$, as observed.

Note that this analysis does not apply to the setting in which the ratio $N/n$ is bounded. We find, accordingly, that this decay is unobserved in our experiments in which the number of labeled samples is in constant proportion with the budget.

# F    Proofs

Unless explicitly stated otherwise, we prove results for the generalized setup outlined in Section B.

## F.1    Proof of Theorem 2

For $\Sigma \in \mathbb{R}^{k \times k}$ symmetric positive-definite, let $\mathcal{P}_\Sigma$ denote the set of distributions on $\mathbb{R}^k$ with covariance $\Sigma$. For a fixed collection of index subsets $\mathcal{I}$ with associated costs $c_I$, let $\Theta_B$ denote the set of budget satisfying estimators $\hat{\theta}$, i.e. the estimators $\hat{\theta}$ which are measurable functions of $n_I$ independent copies of $X_I = (X_i)_{i \in I}$, for each $I \in \mathcal{I}$, such that $\mathbf{B}(\underline{n})$ holds. Note that here we allow for multiple budget inequalities, as described in Section B, and so $\mathbf{B}(\underline{n})$ denotes the proposition that all budget constraints are satisfied simultaneously. We emphasize that we make no explicit restriction to linear or unbiased estimators.

**Theorem 12** (Minimax optimality for general budget constraints)**.**  *We have*

$$\inf_{\hat{\theta} \in \Theta_B} \sup_{P \in \mathcal{P}_\Sigma} \mathbb{E}\left[(\hat{\theta} - \theta^*)^2\right] = \mathrm{Var}\left(\hat{\theta}_{\text{Multi-allocate}(\Sigma)}\right) = \mathcal{V}_B$$

*where the variance is with respect to any distribution $P \in \mathcal{P}_\Sigma$.*

*Proof.*  We first reduce to the case of known and fixed $\underline{n}$.

**Lemma 13.**  *Let $\Theta^{(\underline{n})}$ denote the set of measurable functions $\hat{\theta}$ which are functions of $n_I$ independent copies of $X_I$, for each $I \in \mathcal{I}$. Then if $\mathrm{supp}(a) \subseteq \bigcup\{I : n_I > 0\}$,*

$$\inf_{\hat{\theta} \in \Theta^{(\underline{n})}} \sup_{P \in \mathcal{P}_\Sigma} \mathbb{E}\left[(\hat{\theta} - \theta^*)^2\right] = \min_{\underline{\lambda}\,:\,\mathbf{U}(\underline{n}, \underline{\lambda})} \sum_{I : n_I > 0} \frac{1}{n_I} \lambda_I^\top \Sigma_I \lambda_I;$$

*otherwise, $\sup_{P \in \mathcal{P}_\Sigma} \mathbb{E}\left[(\hat{\theta} - \theta^*)^2\right]$ is unbounded for all $\hat{\theta} \in \Theta_B$.*

We now reduce the conjecture to this lemma. Observe that

$$\Theta_B = \bigcup_{\underline{n}\,:\,\mathbf{B}(\underline{n})} \Theta^{(n)}$$

and so the left hand-side of the conjecture is equal to

$$\inf_{\underline{n}\,:\,\mathbf{B}(\underline{n})} \inf_{\hat{\theta} \in \Theta^{(\underline{n})}} \sup_{P \in \mathcal{P}_\Sigma} \mathbb{E}\left[(\hat{\theta} - \theta^*)^2\right] = \inf_{\underline{n}\,:\,\mathbf{B}(\underline{n})} \min_{\underline{\lambda}\,:\,\mathbf{U}(\underline{n}, \underline{\lambda})} \sum_{I : n_I > 0} \frac{1}{n_I} \lambda_I^\top \Sigma_I \lambda_I =: \mathrm{Var}(\hat{\theta}_{\text{Multi-allocate}(\Sigma)})$$

since $\mathbf{U}(\underline{n}, \underline{\lambda})$ is feasible for $\underline{\lambda}$ if and only if $\mathrm{supp}(a) \subseteq \bigcup\{I : n_I > 0\}$.

It now suffices to prove the lemma.   $\square$

*Proof of Lemma 13.*  The claim that $\sup_{P \in \mathcal{P}_\Sigma} \mathbb{E}\left[(\hat{\theta} - \theta^*)^2\right]$ is unbounded for all $\hat{\theta} \in \Theta_B$ if $\mathrm{supp}(a) \not\subseteq \bigcup\{I : n_I > 0\}$ follows from the observation that if $i \in \mathrm{supp}(a) \setminus \bigcup\{I : n_I > 0\}$, there exist distributions $P \in \mathcal{P}_\Sigma$ such that $\theta_i^* = \mathbb{E}[X_i]$ may be made arbitrary large, while $\hat{\theta}$ cannot depend on such $X_i$.

Therefore, in what follows, we assume $\mathrm{supp}(a) \subseteq \bigcup\{I : n_I > 0\}$. The upper bound is clear from that fact that

$$\{\hat{\theta}_{\underline{n}, \underline{\lambda}} : \mathbf{U}(\underline{n}, \underline{\lambda})\} \subseteq \Theta^{(\underline{n})}$$

i.e., the set of unbiased linear estimators depending on $\underline{n}$ samples is a subset of the set of all estimators depending on $\underline{n}$ samples; and from the fact that $\mathrm{Var}(\hat{\theta}_{\underline{n}, \underline{\lambda}}) = \sum_{I : n_I > 0} \frac{1}{n_I} \lambda_I^\top \Sigma_I \lambda_I$ for every $P \in \mathcal{P}_\Sigma$, hence the minimal MSE of such estimators is precisely the right-hand side.

We now prove the lower bound. Since the Bayes risk for any prior $\mu$ lower bounds the minimax risk, it suffices to construct a sequence of priors $\mu$ for which the risk of the Bayes estimator tends upward to our claimed lower bound. Let us choose the distribution $X \sim \mathcal{N}(\mu, \Sigma)$, and supply the

prior $\mu \sim \mathcal{N}(0, \tau^2 \mathrm{Id}_k)$ for $\tau > 0$ arbitrary; we will later take $\tau \to \infty$. Note that we then have $X_I = P_I X \sim \mathcal{N}(P_I \mu, P_I \Sigma P_I^\top)$.

By construction, any estimator $\hat{\theta} \in \Theta^{(n)}$ depends on the independent set $\bigcup_{I \in \mathcal{I}} \{X_I^{(j)}\}_{1 \leq j \leq n_I}$ where each $X_I^{(j)}$ is distributed according to $\mathcal{N}(\mu_I, \Sigma_I)$. The posterior[4] is then

$$\mu \ \Bigg| \ \bigcup_{I \in \mathcal{I}} \{X_I^{(j)}\}_{1 \leq j \leq n_I} \sim \mathcal{N}(m_\tau, S_\tau)$$

$$S_\tau = \left( \frac{1}{\tau^2} \mathrm{Id}_k + \sum_{I \in \mathcal{I}} n_I P_I^\top \Sigma_I^{-1} P_I \right)^{-1}$$

$$m_\tau = S_\tau \left( \sum_I n_I P_I^\top \Sigma_I^{-1} \overline{X}_I \right)$$

The Bayes risk of estimating $\theta = a^\top \mu$ is then $a^\top S_\tau a$. Letting $\tau \to \infty$, we have shown that the minimax risk is at least[5]

$$a^\top S a, \quad S = \left( \sum_{I \in \mathcal{I}} n_I P_I^\top \Sigma_I^{-1} P_I \right)^\dagger.$$

It remains to show that this risk is achievable by the $\hat{\theta}_{\underline{n}, \underline{\lambda}}$ for some choice of $\underline{\lambda}$ satisfying $\mathbf{U}(\underline{n}, \underline{\lambda})$. We quickly verify this below:

Putting[6]
$$\lambda_I = \left( n_I \Sigma_I^{-1} P_I \right) S a$$

we see that indeed $\mathbf{U}(\underline{n}, \underline{\lambda})$ holds. Moreover, we calculate

$$\mathrm{Var}(\hat{\theta}_{\underline{n}, \underline{\lambda}}) = \sum_I n_I a^\top S P_I^\top \Sigma_I^{-1} \Sigma_I \Sigma_I^{-1} P_I S a = a^\top S \left( \sum_{I : n_I > 0} n_I P_I^\top \Sigma_I^{-1} P_I \right) S a = a^\top S a$$

as desired. This concludes the proof. $\qquad \square$

### F.2 PROOF OF THEOREM 3

We prove a generalization of Theorem 3 in which we allow for multiple budget inequalities.

Fix a vector $B_0 \in \mathbb{R}_{>0}^m$. We consider the limit in which our budget is $B = t \cdot B_0$ and let $t \to \infty$. Suppose that $\widehat{\Sigma} \xrightarrow{p} \Sigma$ in the operator norm, potentially dependent on the variables sampled $X_I$.

We assume the following condition: Suppose that the following problem has a unique minimizer $\underline{\nu}$:

$$\underline{\nu}^\star = \mathrm{argmin}_{\underline{\nu}} V(\underline{\nu}) := a^\top \left( \sum_I \nu_I P_I^\top \Sigma_I^{-1} P_I \right)^\dagger a$$

$$\text{s.t.} \quad \underline{\nu} \geq 0, \quad \sum_I \nu_I c_I \leq B_0, \quad \mathrm{supp}(a) \subseteq \bigcup \{I : \nu_I > 0\}$$

(14)

**Theorem 14** (Generalized asymptotic normality). *Suppose $X \in \mathbb{R}^k$ has finite second moment, and suppose that $\Sigma = \mathrm{Cov}(X)$ satisfies Equation (14). Suppose that $\widehat{\Sigma} \xrightarrow{p} \Sigma$ in the operator norm as*

---

[4] Morally, we are done at this point: the posterior mean is linear in $(\overline{X}_I)_I$, and the Multi-PPI estimator is the best such linear estimator. However, this does not yet directly imply the result. See next page for calculation of the posterior.

[5] Here we use the assumption that $\mathrm{supp}(a) \subseteq \bigcup \{I : n_I > 0\}$, and thus $a$ lies in the range of $\sum_I n_I P_I^\top \Sigma_I^{-1} P_I$.

[6] To find this choice organically, one may solve an infimal norm convolution with Lagrange multipliers.

$t \to \infty$. Let $\hat{\theta}_{\text{MultiPPI}(\widehat{\Sigma})}$ denote the MultiPPI estimator with budget $tB_0$. Then for $\hat{\theta}_{\text{MultiPPI}(\widehat{\Sigma})}$ arbitrarily dependent on any potential samples used to estimate $\widehat{\Sigma}$, we have

$$\sqrt{t}\left(\hat{\theta}_{\text{MultiPPI}(\widehat{\Sigma})} - \theta^*\right) \xrightarrow{d} \mathcal{N}(0, \mathcal{V}^*)$$

as $B \to \infty$, where $\mathcal{V}^* = \lim_{B \to \infty} B\mathcal{V}_B$, and $\mathcal{V}_B$ is defined in Equation (7).

While $\hat{\theta}_{\text{MultiPPI}(\Sigma)}$ is minimax optimal in the setting of fixed and known covariance $\Sigma$, it is in general not efficient, and the variance $\mathcal{V}$ can in general be improved by slowly concatenating onto $X$ nonlinear functions of its components.

*Proof of Theorem 14.* For fixed $t$, we let allocation $\hat{\underline{n}}$ and weights $\hat{\underline{\lambda}}$ be chosen to minimize the variance of $\hat{\theta} = \hat{\theta}_{\text{MultiPPI}(\widehat{\Sigma})}$ under $\widehat{\Sigma}$ subject to the budget $B = tB_0$, as in the MultiPPI procedure. Since $\widehat{\Sigma} \xrightarrow{p} \Sigma$, Lemma 11 ensures that $\hat{\underline{n}}/t \xrightarrow{p} \underline{\nu}^*$, as defined in Equation (14).

We can write $\sqrt{t}(\hat{\theta} - \theta^*) = \sum_{I \in \mathcal{I}} \hat{\lambda}_I^\top \sqrt{\frac{t}{\hat{n}_I}} W_{I,\hat{n}_I}$, where $W_{I,\hat{n}_I} = \frac{1}{\sqrt{\hat{n}_I}} \sum_{j=1}^{\hat{n}_I} (X_I^{(I,j)} - \mu_I)$. For indices $I$ with $\nu_I^* > 0$, we have $\hat{n}_I \xrightarrow{p} \infty$. Define $n_I^* = \lfloor t\nu_I^* \rfloor$, and let

$$W_I^* := \frac{1}{\sqrt{n_I^*}} \sum_{i=1}^{n_I^*} (X_I^{(I,j)} - \mu_I)$$

It is now enough to show that $W_{I,\hat{n}_I} - W_I^* \xrightarrow{p} 0$, and this will follow from Kolmogorov's inequality. To simplify notation, let us focus on a single subset $I$, and define $Y_j = X_I^{(I,j)} - \mu_I$. Let us also define $S_m = \sum_{j=1}^{m} Y_j$. We must show that

$$\frac{S_{\hat{n}}}{\sqrt{\hat{n}}} - \frac{S_{n^*}}{\sqrt{n^*}} \xrightarrow{p} 0$$

where we have dropped dependence on $I$ for convenience. We decompose

$$\frac{S_{\hat{n}}}{\sqrt{\hat{n}}} - \frac{S_{n^*}}{\sqrt{n^*}} = \underbrace{\frac{S_{\hat{n}} - S_{n^*}}{\sqrt{n^*}}}_{A} + \underbrace{\frac{S_{\hat{n}}}{\sqrt{\hat{n}}}\left(1 - \sqrt{\hat{n}/n^*}\right)}_{B}$$

Fix $0 < \delta < 1$. We work on the event $E_\delta(t) = \{|\hat{n} - n^*| \le \delta t\}$, which holds with high probability.

We first control $A$. On $E_\delta(t)$, $\sqrt{n^*}|A|$ is a sum of at most $\delta t + 1$ i.i.d. copies of $Y_j$. Kolmogorov's inequality then yields

$$\mathbb{P}(A > \epsilon) \le \frac{\delta t + 1}{\epsilon^2 n^*} \le 4\frac{\delta}{\epsilon^2}$$

because $n^* = \lfloor t\nu^* \rfloor$. Taking $\delta \to 0$ yields that $A \xrightarrow{p} 0$.

We next control $B$. Working again on $E_\delta(t)$, we have

$$\frac{S_{\hat{n}}}{\sqrt{\hat{n}}} \le \frac{1}{\sqrt{1 - \delta}}\left(\underbrace{\frac{S_{n^*}}{\sqrt{n^*}}}_{O_p(1)} + \underbrace{\frac{S_{\hat{n}} - S_{n^*}}{\sqrt{n^*}}}_{A}\right)$$

Recognizing the second term as $A \xrightarrow{p} 0$, and the first term as tight by the central limit theorem, we conclude that $S_{\hat{n}}/\sqrt{\hat{n}}$ is tight. Now we conclude that $B \xrightarrow{p} 0$ because $\hat{n}/n^* \xrightarrow{p} 1$.

Having proven $W_{I,\hat{n}_I} - W_I^* \xrightarrow{p} 0$, we conclude that

$$\sqrt{t}(\hat{\theta} - \theta^*) = \sum_{I:\nu_I^* > 0} \frac{1}{n_I^*} \sum_{j=1}^{n_I^*} (\lambda_I^*)^\top (X_I^{(I,j)} - \mu_I) + o_p(1)$$

But this is precisely the desired result, since this is the solution to the continuous optimization problem, and we are done. $\qquad\square$

### F.3 PROOF OF THEOREM 4

*Proof.* The proof is immediate from Theorem 5, proven in Section F.4. □

### F.4 PROOFS FOR SECTION E.1

#### F.4.1 PROOF OF THEOREM 5

*Proof.* We have

$$
\begin{aligned}
R(\widehat{\underline{n}}, \widehat{\underline{\lambda}}) - R(\underline{n}^*, \underline{\lambda}^*) &= R(\widehat{\underline{n}}, \widehat{\underline{\lambda}}) - \widehat{R}_N(\widehat{\underline{n}}, \widehat{\underline{\lambda}}) \\
&\quad + \underbrace{\widehat{R}_N(\widehat{\underline{n}}, \widehat{\underline{\lambda}}) - \widehat{R}_N(\underline{n}^*, \underline{\lambda}^*)}_{\leq 0} \\
&\quad + \widehat{R}_N(\underline{n}^*, \underline{\lambda}^*) - R(\underline{n}^*, \underline{\lambda}^*)
\end{aligned}
\tag{15}
$$

and so it suffices to bound $|R(\widehat{\underline{n}}, \widehat{\underline{\lambda}}) - \widehat{R}_N(\widehat{\underline{n}}, \widehat{\underline{\lambda}})|$ and $|\widehat{R}_N(\underline{n}^*, \underline{\lambda}^*) - R(\underline{n}^*, \underline{\lambda}^*)|$. Define

$$
\begin{aligned}
\Delta_N(\underline{n}, \underline{\lambda}) &= |R(\underline{n}, \underline{\lambda}) - \widehat{R}_N(\underline{n}, \underline{\lambda})| \\
&= \left| \sum_{I \in \mathcal{I}: n_I > 0} \frac{1}{n_I} \lambda_I^\top (\Sigma - \widehat{\Sigma}_N) \lambda_I \right| \\
&\leq \|\Sigma - \widehat{\Sigma}_N\| \sum_{I \in \mathcal{I}: n_I > 0} \frac{1}{n_I} \|\lambda_I\|_2^2
\end{aligned}
\tag{16}
$$

Now since $\underline{n}^0, \underline{\lambda}^0$ satisfies **U** and **B**, we have

$$
R(\underline{n}^*, \underline{\lambda}^*) \leq R(\underline{n}^0, \underline{\lambda}^0), \qquad \widehat{R}_N(\widehat{\underline{n}}, \widehat{\underline{\lambda}}) \leq \widehat{R}_N(\underline{n}^0, \underline{\lambda}^0)
$$

from which it follows that

$$
\sigma_1^2 / n_{I^0}^0 \geq \sum_{I \in \mathcal{I}: n_I^* > 0} \frac{1}{n_I^*} (\lambda_I^*)^\top \Sigma (\lambda_I^*) \geq \gamma_{\min}(\Sigma) \sum_{I \in \mathcal{I}: n_I^* > 0} \frac{1}{n_I^*} \|\lambda_I^*\|_2^2
\tag{17}
$$

and similarly

$$
\widehat{\sigma_1^2} / n_{I^0}^0 \geq \sum_{I \in \mathcal{I}: \widehat{n}_I > 0} \frac{1}{\widehat{n}_I} \widehat{\lambda}_I^\top \widehat{\Sigma}_N \widehat{\lambda}_I \geq \gamma_{\min}(\widehat{\Sigma}_N) \sum_{I \in \mathcal{I}: \widehat{n}_I > 0} \frac{1}{\widehat{n}_I} \|\widehat{\lambda}_I\|_2^2,
$$

where $\gamma_{\min}(A)$ denotes the minimum eigenvalue of the matrix $A$. We deduce that

$$
\sum_{I \in \mathcal{I}: n_I^* > 0} \frac{1}{n_I^*} \|\lambda_I^*\|_2^2 \leq \frac{\Sigma_{11}}{n_{I^0}^0 \gamma_{\min}(\Sigma)}
$$

$$
\sum_{I \in \mathcal{I}: \widehat{n}_I > 0} \frac{1}{\widehat{n}_I} \|\widehat{\lambda}_I\|_2^2 \leq \frac{\widehat{\Sigma}_{N,11}}{n_{I^0}^0 (\gamma_{\min}(\Sigma) - \delta)} \leq \frac{\Sigma_{11} + \delta}{n_{I^0}^0 (\gamma_{\min}(\Sigma) - \delta)}
$$

by Weyl's inequality, where we let $\delta = \|\Sigma - \widehat{\Sigma}_N\|$. Coupled with Equation (16), we have

$$
\Delta_N(\underline{n}^*, \underline{\lambda}^*) \leq \delta \frac{\Sigma_{11}}{n_{I^0}^0 \gamma_{\min}(\Sigma)}
$$

$$
\Delta_N(\widehat{\underline{n}}, \widehat{\underline{\lambda}}) \leq \delta \frac{\Sigma_{11} + \delta}{n_{I^0}^0 (\gamma_{\min}(\Sigma) - \delta)}
$$

Taken together with Equation (15) and the definition of $\Delta_N$, we conclude that

$$
R(\widehat{\underline{n}}, \widehat{\underline{\lambda}}) \leq R(\underline{n}^*, \underline{\lambda}^*) + 4 \frac{\delta}{\gamma_{\min}(\Sigma)} \cdot \frac{\sigma_1^2}{n_{I^0}^0}
$$

for all $\delta \leq \gamma_{\min}(\Sigma)/2$. □

### F.4.2 PROOF OF COROLLARY 6

*Proof.* This follows immediately from Theorem 5 and Corollary 6.20 of Wainwright (2019). □

### F.4.3 PROOF OF COROLLARY 7

*Proof.* This follows immediately from Theorem 5 and Theorem 4.7.1 of Vershynin (2018). □

### F.4.4 PROOF OF COROLLARY 8

*Proof.* This follows immediately from the Gershgorin circle theorem, as $\sum_{t \neq s} \mathrm{Cov}(X_t, X_s) \leq \sqrt{\mathrm{Var}(X_t) \mathrm{Var}(X_s)} < c$, and so $\lambda_{\min}(\Sigma)$ is bounded below for all $k$. On the other hand, $\lambda_{\max}(\Sigma)$ is bounded above on account of the same argument and the assumption that $X_i$ are bounded. □

### F.4.5 PROOF OF THEOREM 9

*Proof.* The result follows immediate from Theorem 5 after the following lemma. □

**Lemma 15.** *Suppose that $\Sigma$ is not a multiple of the identity, and that $X \in \mathbb{R}^k$ is sub-Gaussian with proxy $K$. Let $\gamma_{\max}$ denote the maximum eigenvalue of $\Sigma$. Then the Ledoit-Wolf shrinkage estimator $\widehat{\Sigma}_N^{LW}$ satisfies the bound*

$$\mathbb{E}\|\widehat{\Sigma}_N^{LW} - \Sigma\|_{op} \leq \frac{1}{\sqrt{N}} \sqrt{c_1 K^4 \gamma_{\max}^2 k^2 + c_2 K^8 \gamma_{\max} k^3 / a^2}$$

*where $a^2 := \frac{1}{k} \left\| \Sigma - I \cdot \frac{\mathrm{tr}(\Sigma)}{k} \right\|_F^2$.*

*Proof.* Let $\widehat{\Sigma}_N$ denote the empirical covariance matrix. Recall that by definition

$$\widehat{\Sigma}_N^{LW} = (1 - \hat{\delta})\widehat{\Sigma}_N + \hat{\delta}\hat{m}I$$

where $\hat{m} = \mathrm{tr}(\widehat{\Sigma}_N)/k$, and $\hat{\delta} = \hat{b}^2/\hat{d}^2$; we have $b^2 = \mathbb{E}\|\widehat{\Sigma}_N - \Sigma\|_F^2/k$ and $d^2 = a^2 + b^2$, and $\hat{b}$ and $\hat{d}$ are such that $\hat{b} \to b$ and $\hat{d} \to d$ in quartic mean. Our strategy will be to employ the observation that

$$\begin{aligned}
\|\widehat{\Sigma}_N^{LW} - \Sigma\|_F^2 &= \|(1 - \hat{\delta})(\widehat{\Sigma}_N - \Sigma) + \hat{\delta}(\Sigma - mI)\|_F^2 \\
&\leq \left( |1 - \hat{\delta}|\|\widehat{\Sigma}_N - \Sigma\|_F + |\hat{\delta}|\|\Sigma - mI\|_F \right)^2 \\
&\leq 6\|\widehat{\Sigma}_N - \Sigma\|_F^2 + 4\hat{\delta}^2\|\Sigma - mI\|_F^2
\end{aligned}$$

using the coarse bounds that $|1 - \hat{\delta}| \leq 1, |\hat{\delta}| \leq 1$ and $(u + v)^2 \leq 2u^2 + 2v^2$. It therefore suffices to bound $\mathbb{E}\|\widehat{\Sigma}_N - \Sigma\|_F^2$ and $\mathbb{E}\hat{\delta}^2$.

Since $X$ is sub-Gaussian with proxy $K$, $\widehat{\Sigma}_N$ satisfies

$$\mathbb{E}\|\widehat{\Sigma}_N - \Sigma\|_F^2 \lesssim \frac{K^4}{N} \gamma_{\max}(k^2 + k)$$

by Wainwright (2019). This provides a bound on $b^2$; the estimator $\hat{b}$ is (after truncation) a average of $N$ i.i.d. quartic functionals of $X$ of the form $\|XX^\top - \widehat{\Sigma}_N\|_F^2/k$, each of which have finite second-moment bounded by $cK^8\gamma_{\max}^4 k^2$ by the sub-Gaussian assumption.

We conclude that we may bound

$$\mathbb{E}\hat{b}^2 \lesssim \frac{K^4}{N} \gamma_{\max} k.$$

We proceed by cases to bound $\mathbb{E}\hat{\delta}^2$. On the event $\{\hat{d}^2 > a^2/2\}$, we have $\hat{\delta} \leq 2\hat{b}^2/a^2$, so it will suffices to bound the probability that $\{\hat{d}^2 \leq a^2/2\}$. Since $\hat{d}^2$ is again an average of $N$ i.i.d. quartics in $X$, each of which have second moment bounded by $cK^8\gamma_{\max}^4 k^2$, we have

$$\mathbb{E}(\hat{d}^2 - d^2)^2 \lesssim \frac{K^8}{N} \gamma_{\max}^4 p^2$$

We conclude that by Chebyshev's inequality, we have

$$\mathbb{P}(\hat{d}^2 \leq a^2/2) \leq c'' \frac{K^8}{a^4 N} \gamma_{\max}^4 p^2$$

Lastly, since $0 \leq \hat{\delta} \leq 1$ (since $\hat{b}$ is truncated by $\hat{d}$), we conclude that in all cases

$$\hat{\delta}^2 \leq \hat{\delta} \leq \frac{2\hat{b}^2}{a^2} + \mathbb{1}_{\{\hat{d}^2 \leq a^2/2\}}$$

and so

$$\mathbb{E}\hat{\delta}^2 \leq \frac{2}{a^2} \mathbb{E}\hat{b}^2 + \mathbb{P}(\hat{d}^2 \leq a^2/2) \leq \frac{1}{N} \left( c''' K^4 \gamma_{\max}^2 \frac{k}{a^2} + c'''' K^8 \gamma_{\max}^4 \frac{k^2}{a^4} \right)$$

Taken together, we have shown that

$$\mathbb{E}\|\widehat{\Sigma}_N^{LW} - \Sigma\|_F^2 \leq \frac{1}{N} \left[ c_1 K^4 \gamma_{\max}^2 k^2 + c_2 K^8 \gamma_{\max}^4 \frac{k^3}{a^2} \right]$$

as desired. $\qquad \square$

## F.5 PROOFS FOR SECTION E.2

### F.5.1 PROOF OF THEOREM 10

**Note:** For the purpose of this proof only, we slightly change notation, letting $m$ denote the number of labeled samples rather than $n$. This just has the purpose of clarifying the potential conflict with the notation $n_I$.

*Proof.* Let us introduce the notation that $P_I$ is the orthogonal projection onto coordinates $I$, and thus $P_I^\top \lambda_I$ shares its values with $\lambda_I$ on coordinates $I$, and is 0 elsewhere. As a result, note that we have required

$$\sum_{I:n_I>0} P_I^\top \lambda_I = \mu.$$

Now we aim to minimize

$$\frac{1}{m} \left( \sigma_Y^2 - 2\mu^\top \operatorname{Cov} + \mu^\top \Sigma \mu \right) + \sum_{I:n_I>0} \frac{1}{n_I} \lambda_I^\top \Sigma_I \lambda_I$$

or, expanding,

$$V(n, \lambda) := \frac{1}{m} \left( \sigma_Y^2 - 2 \sum_{I:n_I>0} \lambda_I^\top \operatorname{Cov}_I + \sum_{I,J:n_I,n_J>0} \lambda_I^\top \Sigma_{IJ} \lambda_J \right) + \sum_{I:n_I>0} \frac{1}{n_I} \lambda_I^\top \Sigma_I \lambda_I$$

We are interested in minimizing $V(n, \lambda)$ over all $\lambda$ (by which we mean $(\lambda_I)_{I\in\mathcal{I}}$) and $n$ satisfying the budget constraint $\sum_I c_I n_I \leq C$. We will first minimize over $\lambda$ for fixed $n$: define $U(n) := \min_\lambda V(n, \lambda)$. But

$$V(n, \lambda) = \lambda^\top \begin{pmatrix} \left(\frac{1}{m} + \frac{1}{n_{I_1}}\right) \Sigma_{I_1} \mathbb{1}_{n_I>0} & \cdots & \frac{1}{m} \Sigma_{I_1 I_k} \mathbb{1}_{n_{I_1}, n_{I_k}>0} \\ \vdots & \ddots & \vdots \\ \frac{1}{m} \Sigma_{I_k I_1} \mathbb{1}_{n_{I_k}, n_{I_1}>0} & \cdots & \left(\frac{1}{m} + \frac{1}{n_{I_1}}\right) \Sigma_{I_1} \mathbb{1}_{n_I>0} \end{pmatrix} \lambda - 2\lambda^\top \begin{pmatrix} \frac{1}{m} \operatorname{Cov}_{I_1} \mathbb{1}_{n_{I_1}>0} \\ \vdots \\ \frac{1}{m} \operatorname{Cov}_{I_k} \mathbb{1}_{n_{I_k}>0} \end{pmatrix} + \frac{\sigma_Y^2}{m}$$

is a quadratic form in $\lambda$, where we define $\Sigma_{IJ} = (\Sigma_{ij})_{i\in I, j\in J} = P_I \Sigma P_J^\top$. This is of the form

$$V(n, \lambda) = \lambda^\top \left( \frac{1}{m} S_1 + S_2 \right) \lambda - 2\frac{1}{m} \lambda^\top T + d$$

where

$$S_1 = (\Sigma_{IJ} \mathbb{1}_{n_I, n_J>0})_{I,J\in\mathcal{I}}, \qquad S_2 = \texttt{block\_diag} \left( \frac{1}{n_I} \Sigma_I \mathbb{1}_{n_I>0} \right)_{I\in\mathcal{I}}, \qquad T = (\operatorname{Cov}_I \mathbb{1}_{n_I>0})_{I\in\mathcal{I}}$$

and $d$ is constant in $n, \lambda$. It is known that the minimum value of such a quadratic form is

$$U(n) = \min_\lambda V(n, \lambda) = -\frac{1}{m^2} T^\top \left(\frac{1}{m} S_1 + S_2\right)^+ T.$$

This is because $T$ lies in the range of $\frac{1}{m} S_1 + S_2$. To see this, let us introduce the notation that $\mathcal{I}^+ = \{I \in \mathcal{I} : n_I > 0\}$ and let $\mathcal{I}^0$ be its complement. Reorder $\mathcal{I}$ if necessary so that $\mathcal{I}^+$ strictly precedes $\mathcal{I}^0$. Then $\frac{1}{m} S_1 + S_2$ takes the block form

$$\frac{1}{m} \begin{pmatrix} (\Sigma_{IJ})_{I,J \in \mathcal{I}^+} & 0 \\ 0 & 0 \end{pmatrix} + \begin{pmatrix} \texttt{block\_diag}(\Sigma_I/n_I)_{I \in \mathcal{I}^+} & 0 \\ 0 & 0 \end{pmatrix}.$$

Now, both $(\Sigma_{IJ})_{I,J \in \mathcal{I}^+}$ and $\texttt{block\_diag}(\Sigma_I/n_I)_{I \in \mathcal{I}^+}$ are symmetric positive-definite, hence invertible, on the coordinates $\mathcal{I}^+$, and $T$ has support in the span of the coordinates $\mathcal{I}^+$.

Given the block form shown above, we see that

$$\left(\frac{1}{m} S_1 + S_2\right)^+ = \begin{pmatrix} \left(\frac{1}{m}(\Sigma_{IJ})_{I,J \in \mathcal{I}^+} + \texttt{block\_diag}(\Sigma_I/n_I)_{I \in \mathcal{I}^+}\right)^{-1} & 0 \\ 0 & 0 \end{pmatrix}$$

again in the coordinates in which $\mathcal{I}^+$ precedes $\mathcal{I}^0$.

Continuity of the inverse is now enough to conclude that

$$\lim_{m \to 0} m^2 U(n) = -T^\top \texttt{block\_diag}\left(n_I \Sigma_I^{-1}\right)_{I \in \mathcal{I}} T = -\sum_{I \in \mathcal{I}} n_I \, \mathrm{Cov}_I^\top \, \Sigma_I^{-1} \, \mathrm{Cov}_I =: L(n)$$

But now this is a linear function $L(n)$ in $n$. Consider minimizing this in $n$, subject to the (simplex) budget constraint $n_I \geq 0$, $\sum_I c_I n_I \leq C$. The minimum is achieved on a vertex of the simplex, and the minimizer is unique except in the unlikely situation that

$$\frac{\mathrm{Cov}_I^\top \, \Sigma_I^{-1} \, \mathrm{Cov}_I}{c_I} = \text{constant in } I$$

assuming that $\mathrm{Cov}_I \neq 0$ for some $I$.

Now we claim that $m^2 U(n) \to L(n)$ uniformly in $n$ subject to the budget constraint. For this, it suffices to show that

$$\left(\frac{1}{m}(\Sigma_{IJ})_{I,J \in \mathcal{I}^+} + \texttt{block\_diag}(\Sigma_I/n_I)_{I \in \mathcal{I}^+}\right)^{-1} \to \texttt{block\_diag}(\Sigma_I/n_I)_{I \in \mathcal{I}^+}^{-1}$$

in the operator norm, uniformly in $n$. The Woodbury matrix identity implies that the difference is exactly

$$\texttt{block\_diag}(n_I \Sigma_I^{-1})_{I \in \mathcal{I}^+}(I + m \texttt{block\_diag}(\Sigma_I^{-1}/n_I)_{\mathcal{I}^+}(\Sigma_{IJ})_{I,J \in \mathcal{I}^+})^{-1}$$

Now, we have $0 < n_I \leq C/c_I$ for all $I \in \mathcal{I}^+$ by the constraint. The operator norm is submultiplicative, and the first factor is bounded in norm by a constant multiple of $1/\min_I c_I$. Similarly, we have

$$I + m \texttt{block\_diag}(\Sigma_I^{-1}/n_I)_{\mathcal{I}^+}(\Sigma_{IJ})_{I,J \in \mathcal{I}^+} \succ I + \frac{mC}{\min_I c_I} \texttt{block\_diag}(\Sigma_I^{-1})_{\mathcal{I}^+}(\Sigma_{IJ})_{I,J \in \mathcal{I}^+}$$

The operator norm of the right-hand side goes to $\infty$ uniformly in $n$, so the operator norm of its inverse goes to 0 uniformly as well. In conclusion, we have uniform convergence. Therefore, we have

$$n^*(m) := \mathrm{argmin}_n \min_\lambda V(n, \lambda) \xrightarrow[m \to \infty]{} n^*$$

$\square$

## F.6 Proofs for Section E.4

In this section, we prove Lemma 11. The following suffices for most of the proof:

**Lemma 16.** *The map $F : K \times \mathbf{S}_{++}^k \longrightarrow \mathbb{R} \cup \{\infty\}$ is continuous.*

From this, we have the following:

**Lemma 17.** *The correspondence $\underline{\nu}^*$ is upper hemi-continuous. Consequently, if $\underline{\nu}^*(A^*) = \{\underline{\nu}^0\}$ is a singleton and $A^{(N)} \to A^*$, then for every $\epsilon > 0$ there is some $N_0$ so that $\|\underline{\nu} - \underline{\nu}^0\| < \epsilon$ whenever $\underline{\nu} \in \underline{\nu}^*(A^{(N)})$ and $N \geq N_0$.*

*Proof of Lemma 16.* We need three warm-up lemmas.

**Lemma 18.** *Suppose that, for $I \in \mathcal{J}$, $M_I \in \mathbb{R}^{k \times k}$ is a symmetric PSD matrix which is SPD on the subspace $\mathbb{R}^I$, and zero off of the coordinates. Then $\mathrm{range}(\sum_{I \in \mathcal{I}} M_I) = \mathbb{R}^{\cup \mathcal{J}}$.*

*Proof.* It is clear that $\sum_I M_I$ is SPD on $\bigcup \mathcal{I}$; since it is symmetric, it is invertible on $\bigcup \mathcal{I}$. □

**Lemma 19.** *For all $\epsilon$ small enough, $\mathrm{range}(\sum_I \nu_I^*(A_I^*)^\dagger) \subseteq \mathrm{range}(\sum_I \nu_I A_I^\dagger)$.*

*Proof.* Let us take $\epsilon$ small enough that $\nu_I \geq \nu_I^*/2 > 0$ whenever $\nu_I^* > 0$. Let us also take $\epsilon$ small enough that $A \succ 0$ by Weyl's inequality. Then each $A_I^\dagger$ and $(A_I^*)^\dagger$ is symmetric PSD and SPD on the subspace $\mathbb{R}^I$, and zero off the coordinates.

It now follows from Lemma 18 that $\mathrm{range}(\sum_I \nu_I^*(A_I^*)^\dagger) = \mathrm{span}(e_i : i \in \bigcup_{I:\nu_I^*>0} I) \subseteq \mathrm{span}(e_i : i \in \bigcup_{I:\nu_I>0} I) = \mathrm{range}(\sum_I \nu_I A_I^\dagger)$. □

It follows from the assumption and this lemma that $a \in \mathrm{range}(\sum_I \nu_I A_I^\dagger)$.

We have two final lemmas:

**Lemma 20.** *Suppose that $M$ is symmetric PSD and that $\mathrm{range}(M) = span(J)$. Then we have*

$$x^\top M^\dagger x = x_{(J)}^\top M_{(J)}^{-1} x_{(J)}$$

*where $x_{(J)}$ denotes the restriction of $x$ to the coordinates $J$, and $M_{(J)}$ denotes the restriction of $M$ to the coordinates $J \times J$.*

*Proof.* Since $My \in span(J)$ for all $y \in \mathbb{R}^k$, we have $(My)_i = 0$ whenever $i \in J^c$. Thus every row of $M$ indexed by an element of $J^c$ is zero. By symmetry, the same is true for every column. Thus $M$ takes the block form

$$M = \begin{pmatrix} M_{(J)} & 0 \\ 0 & 0 \end{pmatrix}$$

in the ordered coordinates such that $J$ precedes $J^c$. Now the fact that $M$ is symmetric means that it is invertible on its range, and so $M_{(J)}$ is invertible, and we are done. □

Our last lemma is this:

**Lemma 21.** *Assume that $M, \Xi \in \mathbb{R}^{\ell \times \ell}$ are symmetric PSD, and suppose that $\mathrm{range}(M) = \mathbb{R}^J$. Assume that $\|\Xi\| \leq \lambda_{\min}(M_{(J)})/2$ and $M + \Xi$ is invertible. Then if $x \in \mathrm{range}(M)$, we have*

$$\left| x^\top (M + \Xi)^{-1} x - x_{(J)}^\top M_{(J)}^{-1} x_{(J)} \right| \leq 2\|M_{(J)}^{-1}\|^2 \|\Xi\| \|x\|^2.$$

*On the other hand, if $x \notin \mathrm{range}(M)$, we have*

$$\left| x^\top (M + \Xi)^{-1} x \right| \geq \frac{1}{2\|\Xi\|} \|x_{(J^c)}\|^2.$$

*Proof.* In block form, again in ordered coordinates such that $J$ precedes $J^c$, we have

$$x^\top (M + \Xi)^{-1} x = \begin{pmatrix} x_{(J)} \\ 0 \end{pmatrix}^\top \begin{pmatrix} M_{(J,J)} + \Xi_{(J,J)} & \Xi_{(J,J^c)} \\ \Xi_{J^c,J} & \Xi_{J^c,J^c} \end{pmatrix}^{-1} \begin{pmatrix} x_{(J)} \\ 0 \end{pmatrix} = x_{(J)}^\top \left[ M_{(J)} + S \right]^{-1} x_{(J)}$$

where $S = \Xi/\Xi_{(J^c,J^c)}$ denotes the Schur complement. The norm of the Schur complement is bounded $\|S\| \le \|\Xi\|$, as a result of the forthcoming lemma. By submultiplicativity, we have

$$\left\| (M_{(J)} + S)^{-1} - M_{(J)}^{-1} \right\| \le \left\| M_{(J)}^{-1} \right\| \|S\| \|(M_{(J)} + S)^{-1}\|$$

Now the minimum eigenvalue of $M_{(J)} + S$ is at least $\lambda_{\min}(M_{(J)}) - \|S\| \ge \lambda_{\min}(M_{(J)}) - \|\Xi\| \ge \lambda_{\min}(M_{(J)})/2 = \|M_{(J)}^{-1}\|^{-1}/2$ by Weyl's inequality, and so we conclude that

$$\left\| (M_{(J)} + S)^{-1} - M_{(J)}^{-1} \right\| \le 2\|M_{(J)}^{-1}\|^2 \|\Xi\|$$

as desired, and we are done.

On the other hand, by standard properties of the Schur complement, we have

$$x^\top (M + \Xi)^{-1} x = \begin{pmatrix} x_{(J)} \\ x_{(J^c)} \end{pmatrix}^\top \begin{pmatrix} M_{(J,J)} + \Xi_{(J,J)} & \Xi_{(J,J^c)} \\ \Xi_{J^c,J} & \Xi_{J^c,J^c} \end{pmatrix}^{-1} \begin{pmatrix} x_{(J)} \\ x_{(J^c)} \end{pmatrix} \ge x_{(J^c)}^\top (S')^{-1} x_{(J^c)}$$

where

$$S' = (M + \Xi)/(M + \Xi)_{(J^c,J^c)} = \Xi_{(J^c,J^c)} - \Xi_{(J,J^c)}(M_{(J,J)} + \Xi_{(J,J)})^{-1} \Xi_{(J^c,J)}$$

denotes the Schur complement. We proceed to bound $\|S'\|$:

$$\|S'\| \le \|\Xi\| + \|\Xi\|^2 \|(M_{(J,J)} + \Xi_{(J,J)})^{-1}\| \le \|\Xi\| \left( 1 + \|\Xi\| \cdot 2\|M_{(J)}^{-1}\| \right) \le 2\|\Xi\|$$

since we assume $\|\Xi\| \le \lambda_{\min}(M_{(J)})/2$. We conclude that $\lambda_{\min}((S')^{-1}) \ge 1/(2\|\Xi\|)$, establishing the desired result. $\qquad \square$

Below, we summarize the properties of the Schur complement used above:

**Lemma 22** (Properties of Schur complement). *Let $N$ be a symmetric SPD matrix, and $S$ denote one of its Schur complements. Then we have*

$$x^\top S x \le \begin{pmatrix} x \\ y \end{pmatrix}^\top N \begin{pmatrix} x \\ y \end{pmatrix}$$

*and, if $N$ is invertible,*

$$x^\top S^{-1} x \le \begin{pmatrix} x \\ y \end{pmatrix}^\top N^{-1} \begin{pmatrix} x \\ y \end{pmatrix}$$

*for all $x, y$, whence $\|S\| \le \|N\|$.*

*Proof.* The proof is immediate from equation 7.7.5. of Horn & Johnson (2012), using the fact that the invertibility of $N$ implies that of $S$ by Schur's formula. $\qquad \square$

Now we conclude the proof. Let $I_*^+$ denote $\bigcup_{I:\nu_I^* > 0} I$ and $I^+$ denote $\bigcup_{I:\nu_I > 0} I$. By Lemma 18 and Lemma 19 we have $\mathrm{range}(\sum_I \nu_I^*(A_I^*)^\dagger) = \mathbb{R}^{I_*^+} \subseteq \mathbb{R}^{I^+} = \mathrm{range}(\sum_I \nu_I(A_I)^\dagger)$. By assumption, we have $a \in \mathbb{R}^{I_*^+}$.

Applying Lemma 20 to $F$, we have

$$F(\underline{\nu}, A) = a^\top \left( \sum_I \nu_I A_I^\dagger \right)^\dagger a = a_{(I^+)}^\top \left( \left( \sum_I \nu_I A_I^\dagger \right)_{(I^+)} \right)^{-1} a_{(I^+)}$$

Now we write

$$\left(\sum_I \nu_I A_I^\dagger\right) = \left(\sum_I \nu_I^*(A_I^*)^\dagger\right) + \Xi$$

whence

$$\left(\sum_I \nu_I A_I^\dagger\right)_{(I^+)} = \left(\sum_I \nu_I^*(A_I^*)^\dagger\right)_{(I^+)} + \Xi_{(I^+)}$$

with $\|\Xi_{(I^+)}\| \le \|\Xi\|$. Now, the first term is symmetric PSD with range $\mathbb{R}^{I_*^+}$, since $I_*^+ \subseteq I^+$. Thus by Lemma 21 we conclude

$$|F(\underline{\nu}, A) - F(\underline{\nu}^*, A^*)| \le 2 \left\|\left(\sum_I \nu_I^*(A_I^*)^\dagger\right)^{-1}_{(I_*^+)}\right\|^2 \|\Xi\|\|a\|^2$$

and we are done. $\qquad\qquad\square$

### F.6.1 Proof of Lemma 17

*Proof.* The result follows immediately from Berge's theorem after an appropriate compactification of the domain. Let $\tilde{F}(\underline{\nu}, A)$ represent the composition of a homeomorphism $\mathbb{R}_{\ge 0} \longrightarrow \mathbb{R}_{\ge 0} \cup \{\infty\}$ with $F$, e.g. $\tilde{F}(\underline{\nu}, A) = 1 - \exp\{-F(\underline{\nu}, A)\}$. We may then alternatively write

$$\underline{\nu}^*(A) = \operatorname{argmin}_{\underline{\nu} \in K} \tilde{F}(\underline{\nu}, A)$$

where, importantly, $\tilde{F}(\underline{\nu}, A)$ is now real-valued.

It remains to show that $\tilde{F}$ is jointly continuous in $(\underline{\nu}, A)$; Berge's theorem will then imply the desired result. This is equivalent to showing that $F$ is continuous, and so Lemma 16 suffices. $\qquad\square$

Finally, we prove the required lemma regarding convergence of the discrete problem to the continuous one.

*Proof of Lemma 11.* Suppose to the contrary that there is some sequence $N_\alpha, t_\alpha$ of arbitrarily large $N, t$ so that

$$\left\|\frac{\underline{n}}{t_\alpha} - \underline{\nu}^0\right\| \ge \epsilon$$

for some $\underline{n} \in \underline{n}_{t_\alpha}^*(A^{(N_\alpha)})$, for each $\alpha$. By the preceding lemma, we know that there is some $N_0$ so that we may assume $\|\underline{\nu} - \underline{\nu}^0\| < \epsilon/2$ whenever $N \ge N_0$, for all $\underline{\nu} \in \underline{\nu}^*(A^{(N)})$.

For large enough $\alpha$, therefore, we have $\|\underline{\nu} - \underline{\nu}^0\| < \epsilon/2$ for all $\underline{\nu} \in \underline{\nu}^*(A^{(N_\alpha)})$. By the reverse triangle inequality, then, we must have

$$\left\|\frac{\underline{n}}{t_\alpha} - \underline{\nu}\right\| \ge \epsilon/2$$

for all large $\alpha$, for each $\underline{n} \in \underline{n}_{t_\alpha}^*(A^{(N_\alpha)})$ and $\underline{\nu} \in \underline{\nu}^*(A^{(N_\alpha)})$.

Since every such $\underline{n}/t_\alpha$ is contained in the compact set $K$, we may find a limit point $\underline{\nu}' \in K$ so that for all $\delta > 0$, there is some $\alpha$ so that $\|\underline{n}/t_\alpha - \underline{\nu}'\| < \delta$ for some $\underline{n} \in \underline{n}_{t_\alpha}^*(A^{(N_\alpha)})$. For this $x'$ we must then also have

$$\|\underline{\nu}' - \underline{\nu}\| \ge \epsilon/2$$

for each $\underline{\nu} \in \underline{\nu}^*(A^{(N_\alpha)})$.

We now argue that in fact $\underline{\nu}' \in \underline{\nu}^*(A^*)$, which will serve to contradict the assumption that $\underline{\nu}^*(A^*)$ is a singleton. Recall that every $\underline{n} \in \underline{n}_{t_\alpha}^*(A^{(N_\alpha)})$ is optimal, and therefore minimizes $F(\underline{n}, A^{(N_\alpha)})$ over the set $K_t \cap \mathbb{Z}^{|\mathcal{I}|}$, where

$$K_t := \left\{\underline{\nu} : \underline{\nu} \ge 0, \sum_I \nu_I c_I \le t B_0\right\}$$

Since we have $F(\kappa\underline{\nu}, A) = \frac{1}{\kappa}F(\underline{\nu}, A)$ for all $\kappa > 0$, it is not hard to see that if $\underline{\nu}$ is a minimizer of $F(\cdot, A)$ over $K_t$, then $\underline{\nu}/t$ is a minimizer of $F(\cdot, A)$ over $K$. Let us define

$$\underline{n}^*_{t,\text{round}}(A) = \text{round}(t\underline{\nu}^*(A))$$

in the set-valued sense. It follows that $\underline{n}^*_{t,\text{round}}(A) \subseteq K_t \cap \mathbb{Z}^{|\mathcal{I}|}$ for all $A$, and so

$$F(\underline{n}, A^{(N_\alpha)}) \leq F(\underline{n}_{\text{round}}, A^{(N_\alpha)})$$

for all $\underline{n} \in \underline{n}^*_{t_\alpha}(A^{(N_\alpha)})$ and $\underline{n}_{\text{round}} \in \underline{n}^*_{t,\text{round}}(A^{(N_\alpha)})$. By definition, it follows that

$$\underline{n}^*_{t_\alpha,\text{round}}(A^{(N_\alpha)})/t_\alpha \to \underline{\nu}^0$$

in the set valued sense, and so by continuity of $F$ it follows that

$$t_\alpha F(\underline{n}_{\text{round}}, A^{(N_\alpha)}) = F(\underline{n}_{\text{round}}/t_\alpha, A^{(N_\alpha)}) \to F(\underline{\nu}^0, A^*).$$

We conclude that

$$t_\alpha F(\underline{n}, A^{(N_\alpha)}) = F(\underline{n}/t_\alpha, A^{(N_\alpha)}) \leq F(\underline{\nu}^0, A^*)$$

and so taking the limit along the sequence of $\alpha$ previously specified we conclude that

$$F(\underline{\nu}', A^*) \leq F(\underline{\nu}^0, A^*)$$

by continuity of $F$ again. Thus $\underline{\nu}' \in \underline{\nu}^*(A^*)$, yet $\underline{\nu}'$ is separated from $\underline{\nu}^0$ by at least $\epsilon/2$, contradicting the assumption that $\underline{\nu}^*(A^*)$ is a singleton. This concludes the proof.

$\square$

## G    COMPUTATIONAL CONSIDERATIONS

Here we show that the Multi-Allocate procedure reduces to a SOCP in the case of a single budget constraint, and to an SDP in the general case. The proof of Proposition 23 shows that the minimization problem over $\underline{n}, \underline{\lambda}$ may be reduced to one only over $\underline{\lambda}$ via the Cauchy-Schwartz inequality. This minimization over $\underline{\lambda}$ is the dual of an SOCP, as shown by Proposition 24, and the KKT conditions hold. This is

$$\sup_x a^\top x$$

where the supremum is taken over all $x \in \mathbb{R}^k$ such that $x_I^\top \Sigma_I x_I \leq c_I^{-1}$ for all $I \in \mathcal{I}$. This SOCP is simple to implement in the Python package cvxpy.

In the general case, Theorem 2 shows that the optimal choice of $\underline{n}$ is

$$\text{argmin}_{\underline{n} : \mathbf{B}(\underline{n})} a^\top \left( \sum_{I \in \mathcal{I}} n_I P_I^\top \Sigma_I^{-1} P_I \right)^\dagger a$$

Let us denote

$$M(\underline{n}) = \sum_{I \in \mathcal{I}} n_I P_I^\top \Sigma_I^{-1} P_I$$

so that our goal is to solve

$$\min t$$

subject to the constraints that

$$a^\top M(\underline{n})^\dagger a \geq t$$

and $\mathbf{B}(\underline{n})$, which denotes a set of linear constraints on $\underline{n}$. But this is equivalent to the SDP

$$\min t$$

subject to the constraint that

$$\begin{pmatrix} M(\underline{n}) & a \\ a^\top & t \end{pmatrix} \succeq 0$$

and linear constraints on $\underline{n}$. Once again, this is straightforward to implement in cvxpy.

# H  THE DUAL PROBLEM

We briefly recall the setup. Let $\Sigma \in \mathbb{R}^{k \times k}$ be SPD, let $\mathcal{I}$ denote a collection of index subsets $I \subseteq \{1, \ldots, k\}$, and let $c_I$ be a positive scalar defined for every $I \in \mathcal{I}$. It will be convenient to define, for every $I \in \mathcal{I}$, a vector $\lambda_I \in \mathbb{R}^{|I|}$. We denote the concatenation of such vectors by $\underline{\lambda} \in \Lambda = \prod_{I \in \mathcal{I}} \mathbb{R}^{|I|}$. We further recall that $P_I : \mathbb{R}^k \longrightarrow \mathbb{R}^{|I|}$ is the orthogonal projection onto the coordinates indexed by $I$, and set $\Sigma_I = P_I \Sigma P_I^\top$. We define the norm $\|v\|_{\Sigma_I} = \sqrt{v^\top \Sigma_I v}$ on $\mathbb{R}^{|I|}$; this induces the seminorms $\|y\|_{\Sigma_I} = \|P_I y\|_{\Sigma_I}$ on $\mathbb{R}^k$, and $\|\underline{\lambda}\|_{\Sigma_I} = \|\lambda_I\|_{\Sigma_I}$ on $\Lambda$. Lastly, we employ

$$A : \Lambda \to \mathbb{R}^k, \quad A(\underline{\lambda}) = \sum_{I \in \mathcal{I}} P_I^\top \lambda_I$$

to enforce the linear (unbiasedness) constraint $A(\underline{\lambda}) = a$, for some fixed $a \neq 0 \in \mathbb{R}^k$.

Our first step will be to show how to alleviate the budget constraint. To do so, we first briefly recall this constraint. To describe the budget, recall that we define $\underline{n} = (n_I)_{I \in \mathcal{I}} \in \mathbb{Z}_{\geq 0}^{|\mathcal{I}|}$, and employ a budget constraint of the form $\sum_{I \in \mathcal{I}} n_I c_I \leq B$ for a fixed $B > 0$. Denoting $\underline{c} = (c_I)_{I \in \mathcal{I}} \in \mathbb{R}_{>0}^{|\mathcal{I}|}$, our budget constraint may be written $\underline{c}^\top \underline{n} \leq B$. With all of this said, recall that our original problem of interest is

$$V(a) = \min_{\underline{n}, \underline{\lambda}} \sum_{I \in \mathcal{I} : n_I > 0} \frac{1}{n_I} \lambda_I^\top \Sigma_I \lambda_I \qquad \text{s.t.} \sum_{I \in \mathcal{I} : n_I > 0} P_I^\top \lambda_I = a, \quad \underline{c}^\top \underline{n} \leq B \qquad (18)$$

We begin by deriving tractable methods to solve Equation (18). Let us assume for the moment that $\underline{n} \in \mathbb{R}_{\geq 0}^{|\mathcal{I}|}$; we will later construct the final budget allocation by rounding. Our first step is to remove the dependence on $\underline{n}$: we show that the above problem is equivalent to the following:

$$U(a) = \min_{\underline{\lambda} \in \Lambda} \sum_{I \in \mathcal{I}} \sqrt{c_I} \|\lambda_I\|_{\Sigma_I} \qquad \text{s.t.} \quad A\underline{\lambda} = a \qquad (19)$$

We next show that this is equivalent to the dual problem

$$U(a) = \sup_{y \in \mathbb{R}^k} a^\top y \qquad \text{s.t.} \bigwedge_{I \in \mathcal{I}} \left\{ \|y\|_{\Sigma_I}^2 \leq c_I \right\} \qquad (20)$$

Finally, this is a second order cone program, and can be solved with off-the-shelf tools. After we have shown these things, we describe how to convert solutions to Equation (20) into solutions to Equation (18).

**Proposition 23.** *The problems described in Equation* (18) *and Equation* (19) *yield the same optimum* $V = U^2/B$.

**Proposition 24.** *The problems described in Equation* (19) *and Equation* (20) *yield the same optimum* $U$.

*Proof of proposition 23.* We now begin the proof.

$(2) \leq (3)$**:**  Let $A\underline{\lambda} = a$. Define $\underline{n}$ by[7]

$$n_I := \left( \frac{B}{c_I} \right) \frac{\sqrt{c_I} \|\lambda_I\|_{\Sigma_I}}{\sum_{J \in \mathcal{I}} \sqrt{c_J} \|\lambda_J\|_{\Sigma_J}}$$

It is clear that $\underline{c}^\top \underline{n} = B$ by construction, and we have

$$BV(a) \leq \sum_{I : n_I > 0} \frac{B}{n_I} \lambda_I^\top \Sigma_I \lambda_I = \sum_{I : \lambda_I \neq 0} \sqrt{c_I} \|\lambda_I\|_{\Sigma_I} \sum_J \sqrt{c_J} \|\lambda_J\|_{\Sigma_J} = \left( \sum_{I \in \mathcal{I}} \sqrt{c_I} \|\lambda_I\|_{\Sigma_I} \right)^2$$

---

[7]This is defined as long as $\lambda_J \neq 0$ for some $J$; if this fails then $\underline{\lambda} = 0$ and $A\underline{\lambda} = 0$ yields a contradiction.

| Model collection | Cost |
|---|---|
| Gemini 2.5 Pro | $1.25 |
| Gemini 2.5 Flash | $0.30 |
| Both | $1.55 |

Table 1: Cost structure for experiment 1.

$(3) \leq (2)$: Let $\underline{n}, \underline{\lambda}$ satisfy the constraints of Equation (18). Consider the vectors $\underline{c}^{1/2} \odot \underline{n}^{1/2} = (\sqrt{c_I n_I})_{I \in \mathcal{I}}$ and $\left( \mathbb{1}_{n_I>0} n_I^{-1/2} \|\lambda_I\|_{\Sigma_I} \right)_{I \in \mathcal{I}}$ in $\mathbb{R}^{|\mathcal{I}|}$. The Cauchy-Schwartz inequality yields that the product of their squared norms is

$$\left( \sum_I c_I n_I \right) \left( \sum_{I:n_I>0} \frac{1}{n_I} \|\lambda_I\|_{\Sigma_I}^2 \right) \geq \left( \sum_{I:n_I>0} \sqrt{c_I} \|\lambda_I\|_{\Sigma_I} \right)^2$$

Now let us define $\underline{\tilde{\lambda}}$ by $\tilde{\lambda}_I = \lambda_I$ if $n_I > 0$, and $\tilde{\lambda}_I = 0$ otherwise. Then we have

$$A\underline{\tilde{\lambda}} = \sum_I P_I^\top \tilde{\lambda}_I = \sum_{I:n_I>0} P_I^\top \lambda_I = a$$

by assumption, and

$$U(a)^2 \leq \left( \sum_I \sqrt{c_I} \|\tilde{\lambda}_I\|_{\Sigma_I} \right)^2 = \left( \sum_{I:n_I>0} \sqrt{c_I} \|\lambda_I\|_{\Sigma_I} \right)^2 \leq BV(a)$$

and we are done. $\qquad \square$

**Remark 25.** *Note that in general, many $n_I$ will be zero.*

*Proof of proposition 24.* Let $\iota_{\{a\}}$ denote the indicator $b \mapsto \begin{cases} 0 & a = b \\ \infty & a \neq b \end{cases}$.

Then Equation (19) is alternatively written

$$V(a) = \min_{\underline{\lambda} \in \Lambda} g(\underline{\lambda}) + \iota_{\{a\}}(A\underline{\lambda})$$

where $g(\underline{\lambda}) = \sum_I g_I(\lambda_I)$ and $g_I(\lambda_I) = \sqrt{c_I} \|\lambda_I\|_{\Sigma_I}$. We now apply the Fenchel duality theorem. Note that $\iota_{\{a\}}^*(y) = a^\top y$, and $g^*(A^\top y) = \sum_I g_I^*(P_I^\top y) = \sum_I \iota_{\{\|y_I\|_{\Sigma_I} \leq c_I\}} = \iota_{\bigwedge_I \|y_I\|_{\Sigma_I}^2 \leq c_I}$. $\quad \square$

# I    EXPERIMENTAL DETAILS

Here we detail the experimental setup used. We do so in two parts: first, we explain the details for generating the model predictions $(X_2, \ldots, X_k)$ in each experiment; second, we explain the details for constructing the proposed estimator, $\hat{\theta}_{\text{MultiPPI}}$, and the baselines from such predictions.

## I.1    GENERATING MODEL PREDICTIONS

### I.1.1    EXPERIMENT 1: CHATBOT ARENA

We follow the implementation of Angelopoulos et al. (2025) to request autoratings from Gemini 2.5 Pro and Gemini 2.5 Flash. See section E of Angelopoulos et al. (2025) for implementation details.

### I.1.2    EXPERIMENT 2: PROCESSBENCH

We evaluate our method on 500 samples from the OlympiadBench subset of the ProcessBench dataset (Zheng et al., 2024). Binary labels are determined according to whether or not a process error occurred in the given (problem, attempted solution) pair.

In the following, you will see a math problem and an attempted solution. There may or may not be an error in the attempted solution. Your task is to review the attempted solution and decide whether or not it is correct. Report your answer as "correct" or "incorrect" in \boxed{}.
Problem:

Find the smallest number $n$ such that there exist polynomials $f_{1}, f_{2}, \ldots, f_{n}$ with rational coefficients satisfying
$$
x^{2}+7=f_{1}(x)^{2}+f_{2}(x)^{2}+\cdots+f_{n}(x)^{2} .
$$

Attempted solution:

To find the smallest number $\( n \)$, we start by considering the given equation: $\( x^2 + 7 = f\_1(x)^2 + f\_2(x)^2 + \cdots + f\_n(x)^2 \)$. Notice that $\( x^2 + 7 \)$ is always greater than or equal to 7 for any real value of $\( x \)$.
[…]
Therefore, the smallest number $\( n \)$ is $\(\boxed{4}\)$.

Now decide whether or not the attempted solution is correct. Be sure to report your answer as "correct" or "incorrect" in \boxed{}. For example, if you believe that the attempted solution is correct, then you should respond "\boxed{correct}"; if you believe that the attempted solution is incorrect, then you should respond "\boxed{incorrect}". You must respond in exactly this format and include no other text in your response. If you include any additional text in your response, you will be disqualified.

---

Gemini 2.5 Pro: [*Thinking...*]

## — after $B$ words of thought have been produced —

Gemini 2.5 Pro: So, the answer is: \boxed{correct}.

Figure 16: Prompt used to generate autoratings for Experiment 2.

To generate autoratings, we use Gemini 2.5 Pro and truncate its reasoning process at various check-points. Specifically, using the prompt shown in Figure 16, we instruct the model to think for up to 3,000 tokens but interrupt it and demand an answer after $B$ words of thought have been produced, for $B \in \{125, 250, 375, 500\}$, as described in §5. To elicit a definite judgement at each checkpoint, we provide "So, the answer is:" as the assistant and attempt to extract an answer from the subsequent 20 tokens of output with our template.

### I.1.3   EXPERIMENT 3: BIOGRAPHY FACTUALITY

We consider evaluating the factuality of a set of biographies generated by Gemini 2.5 Pro. We replicate the setting of Du et al. (2023): Gemini 2.5 Pro is asked to generate biographies for 524 computer scientists, and we evaluate the factual consistency of such biographies with a set of grounding facts collected by Du et al. (2023).

More specifically, for every person $p \in \mathcal{P}$, we associate a Gemini-generated biography $b^p$ and a set of collected grounding facts $\mathcal{F}^p = \{f_1^p, \ldots, f_{m_p}^p\}$ about the person. Following Du et al. (2023), we estimate the proportion of *factually consistent pairs* $(b^p, f_i^p)$ of generated biographies $b^p$ with each of the collected grounding facts $f_i^p$. Concretely, given the set of all pairs

$$
\mathcal{S} = \{(b^p, f^p) : p \in \mathcal{P}, f^p \in \mathcal{F}^p\}
$$

we *target* the proportion of factually consistent pairs

$$
\frac{\#\{(b, f) \in \mathcal{S} : (b, f) \text{ is factually consistent}\}}{\#\mathcal{S}}
$$

> Biography of Richard Hamming:
> * Worked on the Manhattan Project, contributing to computations on early computing devices.
> * Joined Bell Labs in 1945, working on relay calculators and early digital computers like the IBM 650.
> * Developed Hamming codes, a fundamental set of error detection and correction codes for digital data.
> * Introduced the concept of Hamming distance, a metric for comparing two binary strings.
> ...

> Fact: Richard Hamming was a mathematician who made contributions in computer engineering and communications.
> Gemini 2.5 Pro judgement: Factually consistent.

> Fact: Hamming worked on the Manhattan Project before joining Bell Telephone Laboratories in 1946
> Gemini 2.5 Pro judgement: Factually inconsistent.

Figure 17: Depiction of biography-fact pairs $(b, f)$ as in Experiment 3. Judgements about factual consistency of $(b, f)$ are made by a language model.

We determine the factual consistency, or lack thereof, of a pair $(b, f)$ by majority voting over 5 independent judgments from Gemini 2.5 Pro with thinking. Du et al. (2023) found that judgments by ChatGPT achieved over 95% agreement with human labelers on a set of 100 samples. This level of agreement is evidently not achieved by certain cheaper models, as we proceed to demonstrate experimentally. In Figure 13, we explore using Gemini 2.0 Flash Lite as an autorater for evaluating the factuality consistency of pairs $(b, f) \in \mathcal{S}$.

To elicit better autoratings from queries to Gemini 2.0 Flash Lite, we bootstrap performance via multi-round debate. For a fixed number of agents $A \in \{1, \ldots, 5\}$, and a fixed number of maximum rounds $R \in \{1, 2\}$, we perform the following procedure:

1. $A$ instances of Flash Lite are independently prompted to consider the factual consistency of pairs $(b, f) \in \mathcal{S}$, and provide an explanation for their reasoning.

2. A "pooler" instance of Flash Lite is then asked to review the pair $(b, f)$ and the responses generated by each of the $A$ other instances, and output a judgment in the form of a single word: yes, no, or uncertain.

   (a) If the pooler outputs "yes" or "no," the judgment is final.
   (b) If the pooler outputs "uncertain" and the number of maximum rounds $R$ has not yet been reached, the $A$ instances of Flash Lite are independently shown their prior responses, and the prior responses of each other, and prompted to continue reasoning given this additional information. This procedure continues until either the pooler no longer reports "uncertain," or the maximum number of rounds $R$ has been reached.
   (c) If the pooler outputs "uncertain" and the maximum number of rounds $R$ has been reached, a fair coin is flipped and "yes" or "no" are reported with equal probability.

Since the dataset is balanced, the outcome described in (c) is fair insofar as it is as good as random guessing. We impose the maximum round restriction to encapsulate our budget constraint. To reduce randomness, we generate all autoratings twice, so that the resulting dataset has an effective size of 1048.

**Target:** Proportion of factually-consistent pairs, $\#\{(b, f) \in \mathcal{S} : (b, f) \text{ is factually consistent}\}/\#\mathcal{S}$

**Model family:** $\{\text{The output of the above procedure given } (A, R) : A \in \{1, \ldots, 5\}, R \in \{1, 2\}\}$

**Cost structure:** For a given $(A, R)$, the cost is $A \cdot R$. For collections of models, the cost is additive.

### I.2 CONSTRUCTING THE MULTIPPI ESTIMATOR

For the results shown in §6, we draw 250 fully-labeled samples from each dataset above. We then follow the procedure described in §C.3 for $N = 250$, using the empirical distribution over each dataset as our source of randomness. In § D.2, we replicate the study over a range of values of $N$.

