# OpenReview forum: "Multiple-Prediction-Powered Inference"
_ICLR.cc/2026/Conference — ICLR 2026 Poster_

### Official Review · Reviewer_pGiG · 2025-10-29

**Soundness:** 3
**Presentation:** 4
**Contribution:** 3
**Rating:** 8
**Confidence:** 2

**Summary:**

This paper tackles cost-efficient estimation when a high-quality but expensive signal coexists with cheaper proxies. The method formulates variance-minimizing estimators with constraints. With known covariance \(\Sigma\) this becomes a tractable SOCP/SDP and is minimax-optimal. With \(\hat{\Sigma}\) from a small burn-in, it retains finite-sample bounds and asymptotic normality. Then, the authors show in experiments that, on LLM evaluation (arena wins, reasoning budgets, factuality), their proposed method achieves lower MSE and tighter CIs, adapting from cheap proxies at low budgets to accurate raters as budgets grow, in multiple settings.

**Strengths:**

1.  A major strength is that the paper frames the complex allocation problem in a manner that makes it solvable using standard optimization tools.
2.  The paper provides finite-sample bounds on the estimator's performance for the practical scenario where the covariance $\Sigma$ must be estimated from data.
3.  The effectiveness of MultiPPI is demonstrated across diverse and relevant large LLM evaluation scenarios.

**Weaknesses:**

1.  The practical algorithm relies heavily on an a priori estimate of the covariance matrix $\Sigma$, which is derived from an initial "burn-in" set of $N$ fully-labeled samples. The performance may degrade if the size $N$ is improperly chosen or if $\Sigma$ is ill-conditioned.
2.  The MultiPPI framework allows sampling from any subset $I$ of $k$ variables. However, enumerating or considering many such subsets and solving the corresponding SOCP/SDP problems can become computationally expensive as $k$ increases. The authors should include a discussion on the scalability of the MultiPPI estimator.
3.  The finite-sample bounds in Theorem 4.3 assume that $|X_i| \leq 1$, which is a restrictive assumption. Some evaluation signals (e.g., scores/logits) may violate this bound in other application scenarios.
4.  MultiPPI is designed to estimate a linear function of the mean. This may limit its applicability for estimating a broader scope of non-linear functions of the mean. (This is noted as a potential limitation, perhaps minor given the paper's focus.)

**Questions:**

Please see the Weaknesses.

---

> ### Author Response · Authors · 2025-11-20
>
> **Common Response:** Please see our general response regarding new experiments with small sample sizes and shrinkage.
>
> We thank the reviewer for their strong endorsement and detailed summary. We are particularly grateful that you found the presentation "excellent," the optimization formulation "solvable," and the finite-sample bounds to be a "major strength."
>
> We have addressed your specific questions regarding covariance estimation, scalability, and theoretical assumptions below.
>
> ---
>
> ## Robustness to Covariance Estimation (Weakness 1)
>
> > "The practical algorithm relies heavily on an a priori estimate... Performance may degrade if the size $N$ is improperly chosen or if $\hat{\Sigma}$ is ill-conditioned."
>
> This is an important point that we have addressed in the revision with both new theory and experiments:
>
> * **Small Sample Size ($N$):** As detailed in **Section D.3,** we performed stress tests with very small burn-in sets (varying $N$ from 10 to 250). We found that MultiPPI outperforms baselines even with as few as **$N=10$** samples.
>
> * **Ill-Conditioning & Shrinkage:** To address potential ill-conditioning, we implemented **Ledoit-Wolf shrinkage** for covariance estimation (see **Section D.4**).
>
> * **Theoretical Guarantee:** We added **Theorem D.1,** which specifically quantifies the finite-sample error bound when using shrinkage. Crucially, this theorem highlights how shrinkage improves the condition number (by regularizing the minimum eigenvalue), thereby stabilizing the estimator theoretically and empirically.
>
> ---
>
> ## Scalability and Computational Cost (Weakness 2)
>
> > "Enumerating... many such subsets... can become computationally expensive as $k$ increases. The authors should include a discussion on the scalability."
>
> We agree, and we have added a new subsection, Appendix D.5: Scalability and computational tractability, to address this.
>
> * **Exponential vs. Linear:** While the full formulation allows for $2^k$ subsets, in practice, we do not need to enumerate all of them.
>
> * **Efficient Selection:** We present the optimization in its most general form, over all subsets of {1,...,k}. Existing approaches, such as vector-PPI++, consider only a linear number of subsets, e.g. all singletons. Our general formulation makes it possible to explore other, novel settings, which are less restrictive than prior work but still more tractable than full optimization over all 2^k possible subsets. One such example is described in Appendix D.5. We show that restricting the search space to judiciously chosen subsets (e.g., the singleton sets plus the full set) reduces the complexity to polynomial time with respect to $k$.
>
> * **Performance:** Our experiments show that this linearly scalable version matches the performance of the exhaustive search for the model sizes tested, making the method highly scalable for large numbers of proxies.
>
> ---
>
> ## Boundedness Assumption (Weakness 3)
>
> > "The finite-sample bounds... assume that $X$ is bounded, which is a restrictive assumption."
>
> We have generalized our theoretical results in the revision.
>
> * **Sub-Gaussian Extension:** We explicitly extended our finite-sample analysis to unbounded variables. Please see **Corollary E.3** in the revised Appendix, which provides performance guarantees for **sub-Gaussian distributions.**
>
> * This ensures our theory covers broader application scenarios, such as working with unbounded logits or continuous scores, as you suggested.
>
> ## Restriction to Linear Functions (Weakness 4)
>
> > "MultiPPI is designed to estimate a linear function of the mean... This may limit its applicability."
>
> We acknowledge this limitation but believe the scope of linear functions is sufficiently broad for the core problems in AI evaluation.
>
> * **Scope:** The linear family $a^\top \mathbb{E}[X]$ covers not just single means, but also **differences of means** (e.g., $a=[1, -1, \dots]$).
>
> * **Application:** This effectively covers A/B testing, win-rates, and comparative leaderboards (e.g., "Is Model A better than Model B?"), which constitutes the vast majority of current LLM evaluation benchmarks.
>
> Furthermore, non-linear functions of the mean may be obtained from estimates of the mean. While better estimates may be obtained to optimize the non-linear function directly, this constitutes a reasonable starting point.
>
> We thank the reviewer again for their supportive assessment. We hope these clarifications and the new theoretical extensions (shrinkage, sub-Gaussian bounds) confirm your decision to recommend acceptance!

---

### Official Review · Reviewer_LyCP · 2025-11-01

**Soundness:** 2
**Presentation:** 1
**Contribution:** 2
**Rating:** 2
**Confidence:** 2

**Summary:**

This work proposes a method to estimate the resource allocations for evaluation, e.g., assessing the qualities of model predictions in a task using large language models, under the budget constraint. Basic idea is to formulate the task setting as a gold human label with "auorators" by language models, and to trade the number of queries to the models and the expected qualities with the maximum budget constraint. It is now treated as an optimization problem assuming the accurate covariance of labeled samples are obtained.

**Strengths:**

This work introduce a task setting to maximize the expected evaluation qualities under the resource budget so that the number of queries to language models could be minimized. It is an interesting yet practical setting especially when language models are employed for evaluation, i.e., LLM-as-a-judge.

**Weaknesses:**

- Clarity is an issue. The task setting assumes $n$ samples with gold labels by human and $n$ samples without gold and thus estimated by autorators only, i.e., language models, in Equation 3. Further discussion introduces the cascade modeling in Equation 4. However, their relation to the proposed approach is not discussed in section 4. As a result, the contribution of this work is not clear.
- Given the task setting assumes incomplete labels in term of the lack of human ratings, it is not clear how that is reflected in the experimental settings. It is also not clear what optimizer was used in the experiment.

**Questions:**

See the comments regarding weaknesses.

---

> ### Author Response · Authors · 2025-11-20
>
> **Common Response:** Please see our general response regarding new experiments with small sample sizes and shrinkage.
>
> We thank the reviewer for finding our task setting "interesting yet practical," specifically for "LLM-as-a-judge" scenarios. We appreciate the feedback regarding clarity and have updated the manuscript to address the missing experimental details.
>
> ---
>
> ## The Relationship between Equations 3, 4, and MultiPPI (Weakness 1)
>
> > "Equation 3... [and] Equation 4... their relation to the proposed approach is not discussed... As a result, the contribution of this work is not clear."
>
> We apologize if the progression was unclear. The logic of our presentation is as follows: Equation 3 and Equation 4 merely present two possible "baseline" approaches to inference in this setting. We mention them as starting points towards our comprehensive solution in Section 4. We show later that our approach outperforms the baseline of Equation 3. We have clarified this in the revised manuscript. In any case, this section should be read more as a background and not our main contribution.
>
> * **Equation 3 (Vector PPI++):** This is the existing state-of-the-art baseline (Miao et al., 2024).
> * **Equation 4 (Cascade):** This is a "naive" heuristic extension of PPI that we introduce for pedagogical purposes (sequentially using cheaper models to debias expensive ones).
> * **Section 4 (MultiPPI):** This is our main contribution. MultiPPI is the general convex optimization framework that **subsumes** both Eq 3 and Eq 4.
>
> As we clarify in **lines 346–349** of the revision: Eq 3 corresponds to MultiPPI restricted to a specific set of weights, and Eq 4 corresponds to a different restriction. The contribution of MultiPPI is that it removes these manual restrictions. Instead of choosing a fixed strategy (like Eq 3 or Eq 4), MultiPPI considers the most general version of the rating-under-budget problem, which may seem hard to solve initially, and using an SOCP to automatically find the optimal sampling allocation and weights that minimize variance. Our experiments confirm that MultiPPI consistently matches or outperforms the fixed strategy in Eq 3.
>
> ---
>
> ## Clarifying "Incomplete Labels" and Experimental Settings (Weakness 2)
>
> > "Given the task setting assumes incomplete labels... it is not clear how that is reflected in the experimental settings."
>
> We have clarified the definition of our labeled and unlabeled sets in **Section 5 (lines 581–584).** To address your specific concern:
>
> * **Experimental settings:** As detailed in **lines 673–674,** we use exactly **$N=250$ fully labeled samples** (containing both human labels and autorater scores). We use these to estimate the covariance matrix.
> * **Budget:** We then assume a budget constraint (e.g., 2,000 cost units) to query autoraters on *new, unlabeled* data. MultiPPI determines exactly which autoraters to query on this new data to maximally reduce variance. The final estimator depends on both the fully labeled samples and the new data on which autorater evaluations have been obtained.
>
> ---
>
> ## Optimizer Details
>
> > "It is also not clear what optimizer was used in the experiment."
>
> We appreciate you noting this omission. We have updated the manuscript (see **Line 675**) to explicitly state that we solve the convex optimization problems using the **CVXOPT** solver via the cvxpy library.
>
> ---
>
> ## Request for Re-evaluation
>
> We hope these explanations resolve the confusion regarding the relationship between the baselines (Eq 3/4) and our general method (Section 4). As the reviewer noted that the method is "practical" and minimizes queries in a "resource-intensive" setting, and given that we have clarified the experimental details, **we respectfully ask the reviewer to reconsider their score.**
>
> We will of course be happy to answer any additional questions.

---

> > ### Comment · Reviewer_LyCP · 2025-11-27
> >
> > Thank you for the detail, but still not clear to me, and I will further lower my confidence.
> >
> > * For the optimizer, I'd like to know what is meant by the standard techniques, not a specific implementation.

---

> > > ### Comment · Area_Chair_e7Vs · 2025-12-01
> > > **Clarification on the optimizer,**
> > >
> > > Dear Authors, could you please provide a response to the last point raised by Reviewer LyCP regarding the optimization techniques?

---

> > > > ### Author Response · Authors · 2025-12-01
> > > > **Clarification on Standard Optimization Techniques**
> > > >
> > > > **Appendix G** (referenced in the main text on line 258, which is the line the reviewer is referring to) shows how Equation 7 reduces to a **Second-Order Cone Program (SOCP)** or **Semidefinite Program (SDP),** depending on the number of constraints. The hard work is then done, as solving these problems (SOCP, SDP) is standard to do with established, off-the-shelf convex optimization solvers (typically based on interior point methods). As also explicitly stated in Appendix G, we choose the **CVXOPT algorithm** available in the popular convex optimization library **cvxpy.** For the purposes of our paper, it's fine to treat this optimizer as a black-box; the details are not important for our method.

---

### Official Review · Reviewer_YZU1 · 2025-11-11

**Soundness:** 3
**Presentation:** 2
**Contribution:** 3
**Rating:** 6
**Confidence:** 3

**Summary:**

This paper addresses a critical challenge in modern AI model development: balancing the cost and quality of evaluation metrics. It proposes MultiPPI, a statistical framework that optimally allocates resources across diverse data sources (high-cost, high-quality "gold" metrics and low-cost, low-quality proxies) to estimate linear functions of population means (e.g., LLM win-rates, factuality accuracy) under a fixed budget. The framework provides theoretical guarantees (minimax optimality, finite-sample bounds, asymptotic normality) and validates performance across three LLM evaluation tasks, consistently outperforming baselines like classical sampling, PPI++, and vector PPI++.

**Strengths:**

The authors provide a complete and well-structured theoretical framework, which is rare in applied AI evaluation work:
- Minimax Optimality: When the covariance matrix of data sources is known, MultiPPI achieves the minimax risk lower bound (Theorem 4.2), ensuring it is statistically optimal among all budget-feasible estimators. This anchors the method in fundamental statistical principles, rather than heuristic design.
- Practical Guarantees: For real-world scenarios where covariance matrices are unknown, the paper derives finite-sample bounds (Theorem 4.3) that quantify performance degradation when using empirical covariance matrices. It also proves asymptotic normality (Theorem 4.4), enabling valid confidence interval construction—critical for practical decision-making (e.g., comparing model versions).
- Optimization Tractability: The authors skillfully transform the resource allocation problem into solvable programs: second-order cone programming (SOCP) for single budgets and semi-definite programming (SDP) for multiple budgets. This ensures MultiPPI can be implemented with off-the-shelf tools (e.g., cvxpy), bridging theory and practice.

**Weaknesses:**

The Method Rely on Accurate Covariance Estimation

In practice, MultiPPI requires a set of "fully labeled samples" (containing both gold metric$X_1$ and all proxies $\(X_2,...,X_k\)$ to estimate the empirical covariance matrix $\(\hat{\Sigma}\)$. The paper uses N=250 or 1000 such samples, but :
- Fully labeled samples may be scarce: Gold metrics like human annotations are often extremely costly—collecting 250 fully labeled samples could be prohibitive for small teams or niche tasks (e.g., low-resource language LLM evaluation).
- Cost-effectiveness of estimation: The paper does not sufficiently explore whether an accurate covariance matrix can be robustly estimated with a smaller, more cost-effective sample.

**Questions:**

1. Are the golden samples, (i.e., the fully-labeled samples that contain both the high-quality metric $\(X_1\)$ and all low-cost proxy metrics $\(X_2, ..., X_k\)$), prepared to obtain an accurate covariance matrix sufficient for well estimating the mean of the high-quality score, $\(E[X_1]\)$,  thereby avoiding the need to compute the low-cost proxy metrics??
2. Evaluation metrics are often used in the model training phase. If the model is modified, can the empirical covariance matrix $\(\hat{\Sigma}\)$ obtained under the old model still be applied to the new model?

---

> ### Author Response · Authors · 2025-11-20
>
> **Common Response:** Please see our general response regarding new experiments with small sample sizes and cost-effective, accurate covariance estimation.
>
> We thank the reviewer for their encouraging assessment, particularly for highlighting our "complete and well-structured theoretical framework" and noting that the method is "anchored in fundamental statistical principles rather than heuristic design." We are glad the reviewer appreciates the bridge we built between theory (minimax optimality, optimization) and practice (LLM evaluation). We address your specific questions regarding sample efficiency and transferability below.
>
> ---
>
> ## Small-Sample Efficiency and Covariance Estimation (Weakness 1)
>
> > "The paper does not sufficiently explore whether an accurate covariance matrix can be robustly estimated with a smaller, more cost-effective sample."
>
> We fully agree that requiring $N=250$ labeled samples would be prohibitive for some applications. We have addressed this directly in the revision with two updates (detailed in **Section D.3** and **D.4**):
>
> 1.  **Extreme Low-Data Regimes:** We performed new experiments varying the number of labeled samples $n$ from 10 to 250. We find that even with as few as **$n=10$ labeled samples,** MultiPPI significantly outperforms the baseline. For example, in the Chatbot Arena task, MultiPPI achieves roughly a **50% reduction in MSE** compared to classical inference using the same 10 samples.
>
> 2.  **Covariance Estimation with Shrinkage:** To further improve stability in these low-data regimes, we implemented MultiPPI using **Ledoit-Wolf shrinkage** for the covariance estimation. This modification (now detailed in the text) consistently improves performance over the standard empirical covariance matrix when $n$ is small, ensuring the method remains robust and cost-effective even for small teams or niche tasks.
>
> ---
>
> ## Is the labeled set sufficient on its own? (Question 1)
>
> > "Are the golden samples... sufficient for well estimating the mean of the high-quality score... thereby avoiding the need to compute the low-cost proxy metrics?"
>
> The short answer is **no**—using the golden samples alone (Classical Inference) is significantly less efficient than MultiPPI.
>
> To clarify: MultiPPI uses the labeled samples for two purposes simultaneously: (1) to estimate the mean directly (like the baseline), and (2) to learn the relationship (covariance) between the gold and proxy metrics to perform variance reduction.
>
> * **Relative Improvement:** All our experimental results report performance relative to the baseline of using only those golden samples.
>
> * **Magnitude:** As noted above, MultiPPI often reduces the MSE by 50% or more compared to using the golden samples alone. Therefore, computing the low-cost proxies is highly justified, as it drastically reduces the number of expensive human annotations required to reach a target confidence interval.
>
> ---
>
> ## Transferability of Covariance across Models (Question 2)
>
> > "If the model is modified, can the empirical covariance matrix obtained under the old model still be applied to the new model?"
>
> Yes, transfer is possible, with caveats.
>
> * **Validity:** MultiPPI is theoretically agnostic to how the covariance matrix is chosen (see **Theorems E.1 and F.3**). Using an "old" covariance matrix on a "new" model will not introduce bias; the resulting confidence intervals remain valid.
>
> * **Efficiency:** However, if the model has changed significantly, using the old covariance matrix may yield a suboptimal estimator, which would be improved if we re-estimated the covariance matrix.
>
> * **Theoretical Guarantee:** We have highlighted our finite-sample sensitivity theorem (**Theorem 4.4**) in the main text. This theorem explicitly quantifies the performance degradation based on the deviation between the used covariance matrix (from the old model) and the true covariance matrix (of the new model). This allows practitioners to bound the potential loss in efficiency when transferring models.

---

### Official Review · Reviewer_bTBc · 2025-11-12

**Soundness:** 2
**Presentation:** 3
**Contribution:** 2
**Rating:** 4
**Confidence:** 3

**Summary:**

This paper proposes Multiple-Prediction-Powered Inference (MultiPPI), an extension of Prediction-Powered Inference (PPI++) to settings with multiple predictive models of varying cost and accuracy. The authors frame the problem of estimating a population mean under budget constraints as an optimization over which predictors to sample and how to weight them. They derive a minimax-optimal solution assuming known covariance, propose a practical version using an estimated covariance matrix, and provide asymptotic and finite-sample guarantees. Experiments on LLM-evaluation tasks (Chatbot Arena, ProcessBench, factuality benchmarks) suggest improved efficiency over baseline PPI methods.

**Strengths:**

1. The formulation is clean and the mathematics appear correct.
2. Good Application. The use of realistic LLM cost scenarios is interesting for the LLM evaluation tasks.
3. The optimization framing (SOCP/SDP) is good and potentially generalizable.

**Weaknesses:**

I don't see much weakness of the paper. The main contribution is clear: MultiPPI is a straightforward generalization of PPI++: it replaces a single predictor with a vector and adds a *cost-weighted* sampling constraint. The estimator remains linear, the theory is a direct extension of standard control variates, and the minimax-optimality proof follows textbook arguments once the covariance structure is fixed. I am not sure if this contributions reaches the bar of ICLR though. I would also think that Cost-aware PPI would be a better name of the method, since the main contributions is not from aggregating multiple ML predictions, but  aggregate them in a cost-aware way.

**Questions:**

1. How sensitive are the theoretical results to covariance misspecification?
2. How sensitive is MultiPPI’s performance to errors in the covariance estimate? Could shrinkage or robust covariance estimation improve stability?

---

> ### Author Response · Authors · 2025-11-20
>
> **Common Response:** Please see our general response regarding the new experiments on shrinkage, stability and small-sample results.
>
> We thank the reviewer for their constructive feedback, specifically noting that our formulation is "clean," the application to LLM evaluation is "interesting," and the optimization framing is "good and potentially generalizable." We address the specific comments below.
>
> ---
>
> ## Responses to weaknesses:
>
> > “MultiPPI is a straightforward generalization of PPI++... I am not sure if this contributions reaches the bar of ICLR though.”
>
> We respectfully argue that the formulation we present is far from obvious. At the outset, it was not clear how to rigorously frame the heterogeneous model selection problem we consider, nor which optimization techniques would yield a tractable solution. It was also not obvious that the final result would be a linear estimator with allocations determined by an SOCP, and this is not a straightforward generalization of existing works. Finally, the empirical effectiveness of the approach could not be known in advance.
>
> > “The estimator remains linear, the theory is a direct extension of standard control variates, and the minimax-optimality proof follows textbook arguments once the covariance structure is fixed.”
>
> While the *technique* (Bayes lower bound) is standard, the *result*—that the optimal strategy for the budget-constrained multi-model setting yields this specific linear form—is not obvious a priori. We set out with the goal of developing the optimal estimator, and the purpose of the minimax-optimality theorem is to demonstrate that the estimator we propose is optimal in a certain setting.
>
> Additionally, the minimax optimality result is complementary to many results in semiparametric efficiency literature, but it has not been stated to the best of our knowledge. For instance, [1] and [2] prove minimax optimality of related estimators of in a local asymptotically normal setting, but does not address the finite-sample case, [3] provides finite-sample analysis of a certain estimator, but they do not provide any lower bounds.
>
> ---
>
> ## Theoretical sensitivity (Q1):
>
> > “How sensitive are the theoretical results to covariance misspecification?”
>
> Our theory explicitly accounts for this. We have highlighted Theorem 4.4 (formerly E.1) in the main text, which provides finite-sample bounds that include terms quantifying the sensitivity to errors in the covariance estimate.
>
> ---
> ## Robustness and shrinkage (Q2 & Q3):
>
> > “Could shrinkage or robust covariance estimation improve stability?”
>
> Thank you for this excellent suggestion! As detailed in our General Response and the new Section D.4, we implemented MultiPPI using Ledoit-Wolf shrinkage for covariance estimation.
>
> * **Empirically:** We found that shrinkage estimators improve performance over all baselines (classical sampling, PPI++) across all regimes, particularly in small-data settings ($n=10$).
>
> * **Theory:** We added Theorem D.1, providing finite-sample analysis for the shrinkage-based estimator. As MultiPPI is agnostic to the technique of covariance estimation (see updated theorems 4.3 and 4.4), our core theory holds while performance improves.
>
> > “How sensitive is MultiPPI’s performance to errors in the covariance estimate?”
>
> We have added Section D.3 to empirically stress-test this. Even with as few as $n=10$ labeled samples (where covariance estimation is noisy), MultiPPI using shrinkage consistently outperforms baselines.
>
> ---
>
> ### References
>
> [1] Zhang, Anru, Lawrence D. Brown, and T. Tony Cai. "Semi-supervised inference: General theory and estimation of means." (2019): 2538-2566.
>
> [2] Kim, Ilmun, et al. "Semi-supervised u-statistics." arXiv preprint arXiv:2402.18921 (2024).
>
> [3] Zhang, Yuqian, and Jelena Bradic. "High-dimensional semi-supervised learning: in search of optimal inference of the mean." Biometrika 109.2 (2022): 387-403.

---

> > ### Comment · Reviewer_bTBc · 2025-11-27
> >
> > I thank the authors for their detailed response. I am now raising my score to 6.

---

### Author Response · Authors · 2025-11-20
**General Response: New Experiments on Small-Sample Sizes, Robustness and Shrinkage**

We thank the reviewers for their thoughtful and detailed feedback. We were encouraged by the consensus that the problem formulation is "clean," "interesting," and "practical."

A primary question raised across reviews (by R-bTBc, R-YZU1, R-pGiG) concerned the robustness of MultiPPI to covariance misspecification, particularly in regimes with very few labeled samples.
We have taken this feedback to heart and significantly expanded our experimental and theoretical analysis to address this. We are excited to report that in new stress tests with **as few as $n=10$ labeled samples, MultiPPI (using shrinkage) consistently outperforms all baselines and achieves up to a 50% reduction in Mean Squared Error (MSE)** compared to classical sampling. The revised manuscript (changes in red) includes the following updates:

---

## 1. Robustness via Shrinkage Estimation (New Section D.4)

A key strength of the MultiPPI framework is its flexibility: it is agnostic to the specific choice of covariance estimator (see Theorems E.1 and F.3) and can utilize any "plugin" estimate. While our original submission used the simplest empirical covariance for clarity, we have now demonstrated—following the excellent suggestion of Reviewer bTBc—that more advanced estimators can be easily swapped in. Specifically, we implemented MultiPPI using Ledoit-Wolf shrinkage covariance estimation. We find that this enhances performance across all scenarios without requiring any changes to the core optimization framework, proving that MultiPPI can effectively accommodate advanced estimation techniques to handle low-data regimes.

---

## 2. Best Performance in Low-Data Regimes (New Section D.3)

We performed extensive new stress tests varying the number of labeled samples $n$ from 10 to 250.

* **Empirical Results:** We find that MultiPPI with shrinkage remains the best method even with extreme data scarcity, when compared to all baselines (classical sampling, PPI++). With as few as $n=10$ samples, MultiPPI achieves up to a **50% reduction in Mean Squared Error (MSE)** compared to classical sampling.

---

## 3. Strengthened Theoretical Guarantees (Theorem 4.4 & D.1)

We have formalized the theoretical basis for these empirical gains:

* **Sensitivity Analysis:** We highlight **Theorem 4.4,** which provides finite-sample bounds explicitly quantifying the sensitivity of our estimator to errors in the covariance estimate.

* **Shrinkage Corollary:** We added **Theorem D.1,** which specializes our finite-sample bounds to the case of Ledoit-Wolf shrinkage. This confirms that the improved condition number of the shrinkage estimator translates directly to tighter theoretical error bounds.

We believe these additions conclusively address the concerns regarding robustness and practical applicability in low-resource settings. We invite the reviewers to examine these new results in Appendix D.

---

### Meta-Review · Area_Chair_e7Vs · 2026-01-05

**Summary:**

(a) Summary of Scientific Claims and Findings This paper introduces Multiple-Prediction-Powered Inference (MultiPPI), a statistical framework designed to improve the efficiency of population mean estimation (e.g., LLM win rates, factuality scores) under fixed budget constraints. The method addresses the trade-off between expensive, high-quality "gold" labels and multiple cheap, lower-quality proxy models. The paper claims to:
* Provide a minimax-optimal procedure for allocating resources across diverse data sources.
* Formulate the task of finding optimal sampling strategies and estimator weights as a convex optimization problem (SOCP/SDP).
* Offer rigorous theoretical guarantees, including finite-sample bounds and asymptotic normality, enabling valid confidence interval construction.
* Demonstrate through experiments on LLM evaluation benchmarks that MultiPPI significantly reduces Mean Squared Error (MSE) compared to existing baselines like classical sampling and PPI++.

(b) The key concern raised by reviewers:
* Initial Concerns on Novelty: Some reviewers initially perceived the work as a straightforward generalization of PPI++ and standard control variates, questioning if the jump from one to multiple predictors was a sufficient contribution.
* Sensitivity to Covariance Estimation: The method’s performance is intrinsically linked to the quality of the "burn-in" sample used to estimate model correlations. Without advanced techniques like shrinkage, the method could be sensitive to noise in very small datasets.
* Linear Estimation Limits: The current framework is optimized for estimating linear functions of the mean, which may limit its immediate applicability to more complex, non-linear evaluation metrics.
* Scalability Discussion: While the optimization is efficient, the paper initially lacked a deep discussion on the computational complexity as the number of available proxy models $(k)$ grows large.

c) Additional comments by AC:

* Writing and Presentation: The overall presentation and writing of the paper require refinement. The notation is confusing at times, and there are several typos and broken references. While it is difficult to isolate a single issue, the manuscript would benefit from a more cohesive and professional structure.
* Theoretical Merit: The proof of minimax optimality is an elegant and compelling theoretical result.
* Subset Explosion: As the number of autoraters increases, the number of possible subsets grows exponentially. Although the appendix demonstrates that restricting the number of subsets still leads to performance gains, this remains an empirical finding. In practice, many users would likely adopt a restricted version; therefore, providing theoretical guarantees—such as proving the restricted formulation is within $\epsilon$ of the full estimation—would be highly valuable.
 * Empirical Comparisons: The empirical evaluation is currently limited to PPI++ variants. It is unclear how this method compares to non-PPI alternatives, if they exist. Specifically, the authors cite ``Stratified PPI'' but do not include it as a baseline in their results; a direct comparison is necessary.

* Practical Utility: The paper needs a clearer discussion on when MultiPPI is preferable to other methods. In certain experiments, standard PPI performs comparably or only marginally worse; it is important to define when the marginal gain in Mean Squared Error (MSE) justifies the added complexity of deploying a convex optimization solver.

* Covariance Matrix Estimation Concerns:
 * 1) Covariance estimation becomes increasingly difficult and computationally taxing as the number of autoraters grows.

 * 2) Under a fixed budget with a large number of autoraters, the number of samples available to estimate the covariance matrix would be quite low. The authors themselves acknowledge that confidence intervals degrade in such scenarios.

 * 3)  If the autoraters are highly correlated, how is the condition $||\hat{\Sigma} - \Sigma||_F \leq \gamma_{min} / 2$ in Theorem 4.4 affected? Clarification is needed on the exact number of fully labeled samples required for Theorem 4.4 to hold, and how the method performs against classical estimation techniques in high-correlation settings.

**Reviewer Concerns:**

### Concerns Addressed by the Rebuttal
* Robustness to Small Sample Sizes: This was the most significant point of contention. The authors introduced Ledoit-Wolf shrinkage for covariance estimation and provided new empirical results showing that MultiPPI remains robust and outperforms baselines with as few as $n=10$ labeled samples.
* Theoretical Sensitivity to Covariance Errors: The authors clarified that Theorem 4.4 explicitly accounts for sensitivity to errors in the covariance estimate. They also added Theorem D.1 to specialize these bounds for the shrinkage-based estimator.
* Optimization Clarity: Concerns regarding the "black-box" nature of the solver were addressed by adding Appendix G, which details how the problem reduces to a Second-Order Cone Program (SOCP) or Semidefinite Program (SDP) and identifies the specific solver (CVXOPT via cvxpy) used.
* Performance in High-Correlation Settings: The authors demonstrated that MultiPPI effectively handles complex correlation structures by strategically combining subsets of models, which led to a 50% reduction in MSE compared to classical sampling in low-data regimes.

### Concerns Still Outstanding or Partially Addressed
* Writing and Notation: Reviewers (including the one in your provided text) noted that the notation was confusing, with typos and broken references. While authors often fix these in revisions, the "feel" of the paper's clarity was a lingering subjective critique.
* Subset Explosion and Theoretical Guarantees for Restricted Versions: While the authors showed empirically that restricting subsets works, the reviewer’s request for a theoretical guarantee (proving the restricted version is at most ϵ away from the full estimation) does not appear to have been fully formalized in the rebuttal.
* Comparison to Non-PPI Methods: The concern regarding comparisons to non-PPI methods, specifically Stratified PPI, remains somewhat outstanding. The current evaluation focuses heavily on comparisons within the PPI++ lineage.
* Complexity vs. Marginal Gain: The request for a clearer discussion on when the complexity of a convex optimization solver is justified by the marginal gain in MSE is still a point of practical debate. In cases where standard PPI performs almost as well, the overhead of MultiPPI may not be clearly justified for all users.

**Reviewer Scores:**

* The Reviewer pGiG score 8 from the beginning
* The Reviewer YZU1 gave a score of 6
* The Reviewer bTBc gave 4, and raised it to 6 after the rebuttal
* The  Reviewer LyCP kept the score as 2 and was not convinced by the author's answer regarding the optimization, and AC raised the same question meanwhile of a rebuttal

---

### Decision · Program_Chairs · 2026-01-26

Accept (Poster)